# In mice, discrete odors can selectively promote the neurogenesis of sensory neuron subtypes that they stimulate

Kawsar Hossain[1,2], Madeline Smith[1], Karlin E Rufenacht[1], Rebecca O'Rourke[1], Stephen W Santoro[1]*

[1]Department of Pediatrics, Section of Developmental Biology, University of Colorado School of Medicine, Aurora, United States; [2]Department of Molecular and Cellular Biosciences, University of Cincinnati College of Medicine, Cincinnati, United States

*For correspondence:
Stephen.Santoro@CUAnschutz.edu

Competing interest: The authors declare that no competing interests exist.

## eLife Assessment

This study provides **valuable** insights into the role of sensory stimulation in neurogenesis in the mammalian olfactory epithelium, where new olfactory sensory neurons are continually born throughout an animal's lifespan. The authors show that exposure to two different musk-related odors specifically increases the birth rates of those neurons that respond to these odors. This potentially results in adaptive changes in the subtype composition of the olfactory sensory neuron population. **Solid** evidence, well supported by control experiments, is presented to support these findings, though further work is needed to confirm that this phenomenon generalizes to olfactory sensory neurons expressing other types of odorant receptor and to explore the mechanisms underlying the stimulus specificity of neurogenesis.

**Abstract** In mammals, olfactory sensory neurons (OSNs) are born throughout life, ostensibly solely to replace neurons lost *via* turnover or injury. This assumption follows from the hypothesis that olfactory neurogenesis is stochastic with respect to neuron subtype, as defined by the single odorant receptor that each neural precursor stochastically chooses out of hundreds of possibilities. This assumption is challenged, however, by recent findings that the birthrates of a fraction of OSN subtypes are selectively reduced by olfactory deprivation. These findings raise questions about how, and why, olfactory stimuli are required to accelerate the neurogenesis rates of some subtypes, including whether the stimuli are specific (e.g. discrete odorants) or generic (e.g. broadly activating odors or mechanical stimuli). Based on previous findings that the exposure of mice to sex-specific odors can increase the representations of subtypes responsive to those odors, we hypothesized that the neurogenic stimuli comprise discrete odorants that selectively stimulate OSNs of the same subtypes whose birthrates are accelerated. In support of this, we have found, using scRNA-seq and subtype-specific OSN birthdating, that exposure to male and exogenous musk odors can accelerate the birthrates of subtypes responsive to those odors. These findings reveal that certain odor experiences can selectively 'amplify' specific OSN subtypes and suggest that persistent OSN neurogenesis serves, in part, an adaptive function.

## Introduction

Mammalian olfactory epithelia (OE) contain hundreds of distinct olfactory sensory neuron (OSN) subtypes, each of which expresses a single odorant receptor (OR) and can thereby be stimulated and/or inhibited by a distinct set of odorant molecules (*Imai, 2022*; *Xu et al., 2020*). The olfactory

**eLife digest** Wine tasters and perfumers are living proof that repeated exposure to specific scents can make humans more sensitive to them in the long-term. The sense of smell in humans and other mammals relies on millions of odor-sensing neurons, which detect scent molecules inhaled through the nose and send signals to the brain. There are many different 'subtypes' of these neurons, each of which detects specific odors.

The olfactory epithelium, a tissue lining the nasal cavity, produces odor-sensing neurons through a process called neurogenesis. It has long been assumed that this serves only to replace dead neurons and not to enable changes in the neuron population. This is because the subtype of each new neuron is thought to be random, so should be unaffected by exposure to odors. Yet, a more recent study in mice revealed that blocking one nostril reduces how frequently odor-sensing neurons of some subtypes are generated. However, these experiments did not reveal the type of stimuli that influence neurogenesis in the olfactory epithelium.

To investigate the possibility that these stimuli are specific odors, Hossain et al. exposed mice to odors emitted by male mice or musk odors, which are known to activate specific subtypes of odor-sensing neurons. Genetic and image-based analyses of the olfactory epithelium revealed that this exposure selectively accelerated the generation of neurons that respond to these odors. This suggests that neurogenesis does not just replace 'lost' cells but also allows the olfactory epithelium to adapt to scent exposure.

Hossain et al. hypothesize that this odor-specific neurogenesis could be one mechanism that allows mammals to 'train' their noses to be more sensitive to specific scents. In the future, these findings could help to develop treatments that restore the sense of smell in people who have lost sensory neurons.

---

epithelium is one of a few regions of the mammalian nervous system where neurogenesis occurs throughout life (*Brann and Firestein, 2014*; *Schwob et al., 2017*; *Yu and Wu, 2017*). In the hippo-campus and olfactory bulb, persistent neurogenesis plays vital roles in learning and memory (*Lledo and Valley, 2016*; *Ming and Song, 2011*; *Opendak and Gould, 2015*). By contrast, life-long neurogenesis within the mammalian OE is generally assumed to function solely to replace OSNs that are lost due to normal turnover or environmentally induced damage. This assumption is consistent with the prevailing hypothesis that OSN neurogenesis is strictly stochastic with respect to subtype since it is based on the evidently stochastic process of OR choice (*McClintock, 2015*; *Monahan and Lomvardas, 2015*; *Yusuf and Monahan, 2024*).

Several studies have observed that the relative quantities of distinct OSN subtypes can be altered by manipulating olfactory experience in mice (*Cadiou et al., 2014*; *Cavallin et al., 2010*; *Coppola and Waggener, 2012*; *Dias and Ressler, 2014*; *Fischl et al., 2014*; *Ibarra-Soria et al., 2017*; *Jones et al., 2008*; *van der Linden et al., 2018*; *Morrison et al., 2015*; *Santoro and Dulac, 2012*; *Watt et al., 2004*; *Zhao et al., 2013*; *Vihani et al., 2020*). Olfactory deprivation on one side of the OE via unilateral naris occlusion (UNO), for example, has been found to cause both increases and decreases in the representations of different OSN subtypes on the closed side relative to the open (*Santoro and Dulac, 2012*; *Zhao et al., 2013*). Similar bidirectional changes in subtype representations have been observed following olfactory enrichment *via* exposure to discrete odors in mice (*Ibarra-Soria et al., 2017*; *van der Linden et al., 2018*; *Vihani et al., 2020*). Experience-induced changes in the representations of specific OSN subtypes have generally been attributed entirely to altered OSN survival (*Cadiou et al., 2014*; *Cavallin et al., 2010*; *Jones et al., 2008*; *van der Linden et al., 2018*; *Morrison et al., 2015*; *Santoro and Dulac, 2012*; *Watt et al., 2004*; *Zhao et al., 2013*; *Vihani et al., 2020*; *Ross and Fletcher, 2019*; *Zhao and Reed, 2001*), which could theoretically account for both increases and decreases in subtype representations. By contrast, the stochastic nature of OR choice would seem to preclude the contribution of a mechanism involving subtype-selective changes in OSN birthrates.

Unexplained observations that olfactory deprivation *via* UNO reduces both the overall rate of OSN neurogenesis (*Farbman et al., 1988*; *Cummings and Brunjes, 1994*; *Mirich and Brunjes, 2001*; *van der Linden et al., 2020*) and the representations of a fraction of OSN subtypes (*Coppola and Waggener, 2012*; *Fischl et al., 2014*; *van der Linden et al., 2018*; *Santoro and Dulac, 2012*; *Zhao*

et al., 2013; van der Linden et al., 2020) motivated a recent study to test the possibility that the birthrates of these subtypes depend on olfactory stimulation. To do so, newborn OSNs of subtypes that exhibit altered representations following UNO were quantified on the open side relative to the closed side of OEs from UNO-treated mice (van der Linden et al., 2020). Remarkably, subtypes previously found to have reduced representations on the closed side of the OE also exhibited significantly lower birthrates on the closed side, while subtypes with elevated representations on the closed side showed no differences in birthrates between the two sides. These findings indicated that olfactory stimulation affects the representations of specific OSN subtypes via two distinct mechanisms. One mechanism elevates the representations of specific subtypes via selective increases in their birthrates. By contrast, a second mechanism reduces the representations of subtypes with extremely high levels of baseline activity, presumably via shortened OSN lifespans due to overstimulation (van der Linden et al., 2020).

Findings that the birthrates of a fraction of subtypes depend on olfactory stimulation raise several fascinating questions related to the mechanism and function of this phenomenon. One key question concerns the nature of the stimuli that promote neurogenesis. Because naris occlusion reduces exposure to potentially thousands of odors, as well as mechanical stimuli, and likely causes additional physiological changes (Coppola, 2012), we envisioned that the neurogenic stimuli could be either non-specific with respect to the subtypes that they stimulate (e.g. generic odors, mechanical stimuli, or other physiological effects of UNO) or, alternatively, discrete odorants that selectively stimulate the same OSN subtypes whose birthrates are affected. If the neurogenic stimuli are non-specific, this would imply a generic mechanism and, perhaps, an unknown homeostatic function. By contrast, if the stimuli are discrete odorants that selectively activate the same subtypes whose birthrates are affected, this would imply a highly specific mechanism in which exposure to certain odors can 'amplify' subtypes responsive to those odors.

In this study, we tested the possibility that the neurogenic stimuli comprise discrete odorants. One prediction of this hypothesis is that the extent to which naris occlusion reduces the birthrates of specific subtypes should vary depending on the odor environment to which animals are exposed. Consistent with this, open-side biases in the birthrates of specific subtypes have been found to depend on whether a mouse was in the nursing or post-weaning stage at the time of birthrate assessment (van der Linden et al., 2020). A second prediction is that exposure of intact (non-occluded) mice to specific odors should selectively increase the representations of subtypes responsive to those odors. Support for this comes from two separate studies that identified a set of subtypes that were differentially represented within the OEs of male and female mice that were housed in a sex-separated manner until six months (van der Linden et al., 2018) or up to 43 weeks of age (Vihani et al., 2020), and that these differences were attenuated in mice that were housed sex-combined. Notably, several of these subtypes were found to selectively respond to sex-specific odors (van der Linden et al., 2018; Vihani et al., 2020), indicating that the observed differences were at least partly odor-dependent. As with UNO, the differentially represented subtypes fell into two categories: those for which stimulation increased their representations and those for which stimulation had the opposite effect. OSNs in the former category were exemplified by subtype Olfr235, which exhibited a higher representation in mice exposed to males (van der Linden et al., 2018; Vihani et al., 2020) and selective responsivity to male-emitted odors (van der Linden et al., 2018), and was previously shown to respond to musk odors (McClintock et al., 2014; Sato-Akuhara et al., 2016; Shirasu et al., 2014), whose emission by mice has not been documented. In analogy to findings from UNO-based studies (van der Linden et al., 2020), we speculated that these subtypes undergo sex-specific odor-dependent selective birthrate acceleration. Here, we present evidence that the neurogenesis rates of Olfr235 and other musk-responsive OSN subtypes are increased in mice exposed to male-specific and musk odors. These findings support the hypothesis that discrete odors can selectively accelerate the birthrates of OSN subtypes that they stimulate and suggest that the function of persistent OSN neurogenesis is not limited to the replacement of neurons lost through damage or normal turnover, but may also enable adaptive changes to the subtype composition of the OSN population.

## Results

### Long-term exposure of mice to male odors is associated with increased representations of musk-responsive OSN subtypes

To test the possibility that discrete odorants can selectively accelerate the birthrates of the OSN subtypes that they stimulate, we sought to identify subtypes that exhibit odor exposure-dependent increases in representations and for which stimulating odors have been identified. A challenge of this approach is that odorant ligands remain unidentified for most OSN subtypes (*Peterlin et al., 2014*). To overcome this, we speculated that candidate subtypes might be identified among those previously found to be more highly represented in mice exposed to odors emitted specifically by male or female mice (*van der Linden et al., 2018*; *Vihani et al., 2020*). OSN subtypes in this category, which include Olfr235, were observed to be more highly represented in mice exposed to male conspecifics (sex-separated males; sex-combined males and females) compared to mice isolated from males (sex-separated females; *van der Linden et al., 2018*; *Vihani et al., 2020*; *Appendix 1—figure 1*). These findings, combined with observations that Olfr235 OSNs show selective responsivity to male odors (*van der Linden et al., 2018*), suggest that the exposure of mice to one or more odor components specific to males causes an increase in the representation of subtype Olfr235 within the OSN population. Interestingly, Olfr235, which expresses the OR-encoding gene *Or5an11*, belongs to a group of subtypes that express homologous ORs and respond to different musk odorants with varying levels of sensitivity and selectivity (*McClintock et al., 2014*; *Sato-Akuhara et al., 2016*). Notably, like Olfr235, other subtypes within this group, which includes Olfr1440 (*Or5an6*), Olfr1437 (*Or5an1b*), Olfr1431 (*Or5an9*), and Olfr1434 (*Or5an1*), exhibited higher transcript levels in the OEs of mice exposed to male odors compared to their unexposed counterparts (except subtype Olfr1434, whose transcript levels were too low to be accurately assessed; *van der Linden et al., 2018*; *Appendix 1—figure 1*). Accordingly, RNA-fluorescent in situ hybridization (RNA-FISH) analyses of a subset of these ORs, Olfr235 and Olfr1437, confirmed that the elevated transcript levels observed in mice exposed to male odors reflected greater representations of these subtypes within the OSN population (*van der Linden et al., 2018*). By contrast, subtypes Olfr912 (*Or8b48*) and Olfr1295 (*Or4k45*), which detect the male-specific non-musk odorants 2-(sec-Butyl)–4,5-dihydrothiazole (SBT) and (methylthio)methanethiol (MTMT), respectively (*Vihani et al., 2020*), exhibited lower representations and/or transcript levels in mice exposed to male odors (*van der Linden et al., 2018*; *Vihani et al., 2020*; *Appendix 1—figure 1*), possibly reflecting reduced survival due to overstimulation. Taken together, these findings indicate that OSN subtypes that are responsive to musk odors are selectively increased in their representations upon long-term exposure to male mice, consistent with the hypothesis that components of male odors selectively accelerate the birthrates of these subtypes.

### Olfactory deprivation via UNO reduces quantities of newborn OSNs of musk-responsive subtypes in male mice

If the birthrates of musk-responsive OSN subtypes are accelerated by exposure to male odors, quantities of newborn OSNs of these subtypes would be expected to be reduced following olfactory deprivation in male mice. To test this, we analyzed two single-cell RNA sequencing (scRNA-seq) datasets (one generated previously [OE 1] *van der Linden et al., 2020*) and the other as part of this study [OE 2] corresponding to the open and closed sides of OEs from male mice that had been UNO-treated at postnatal day (PD) 14, weaned sex-separated at PD 21, and dissected at PD 28 (*Figure 1A and B*; *Figure 1—figure supplement 1A*). Within each dataset, newborn OSNs of specific subtypes can be identified based on the co-expression of *Gap43*+ (an established marker of immature OSNs *McIntyre et al., 2010*; *Verhaagen et al., 1989*; *Fletcher et al., 2017*) and specific OR genes (*van der Linden et al., 2020*; *Figure 1C and D*; *Figure 1—figure supplement 1B, C*). Using this approach, we mapped newborn OSNs of each of the five known or putative musk-responsive subtypes within the feature plots corresponding to the open and closed sides of each OE (*Figure 1D*; *Figure 1—figure supplement 1C*) and quantified them as a proportion to total OSNs (*Figure 1E-left*; *Figure 1—figure supplement 1D*) and total cells (*Figure 1—figure supplement 1E*) within the open and closed data subsets. Remarkably, musk-responsive OSN subtypes exhibited, on average, 3.1-fold greater quantities of newborn OSNs within the open-sides of the OEs compared to the closed, with all subtypes except Olfr1437 showing open-side biases (*Figure 1D and E-left*; *Figure 1—figure supplement*

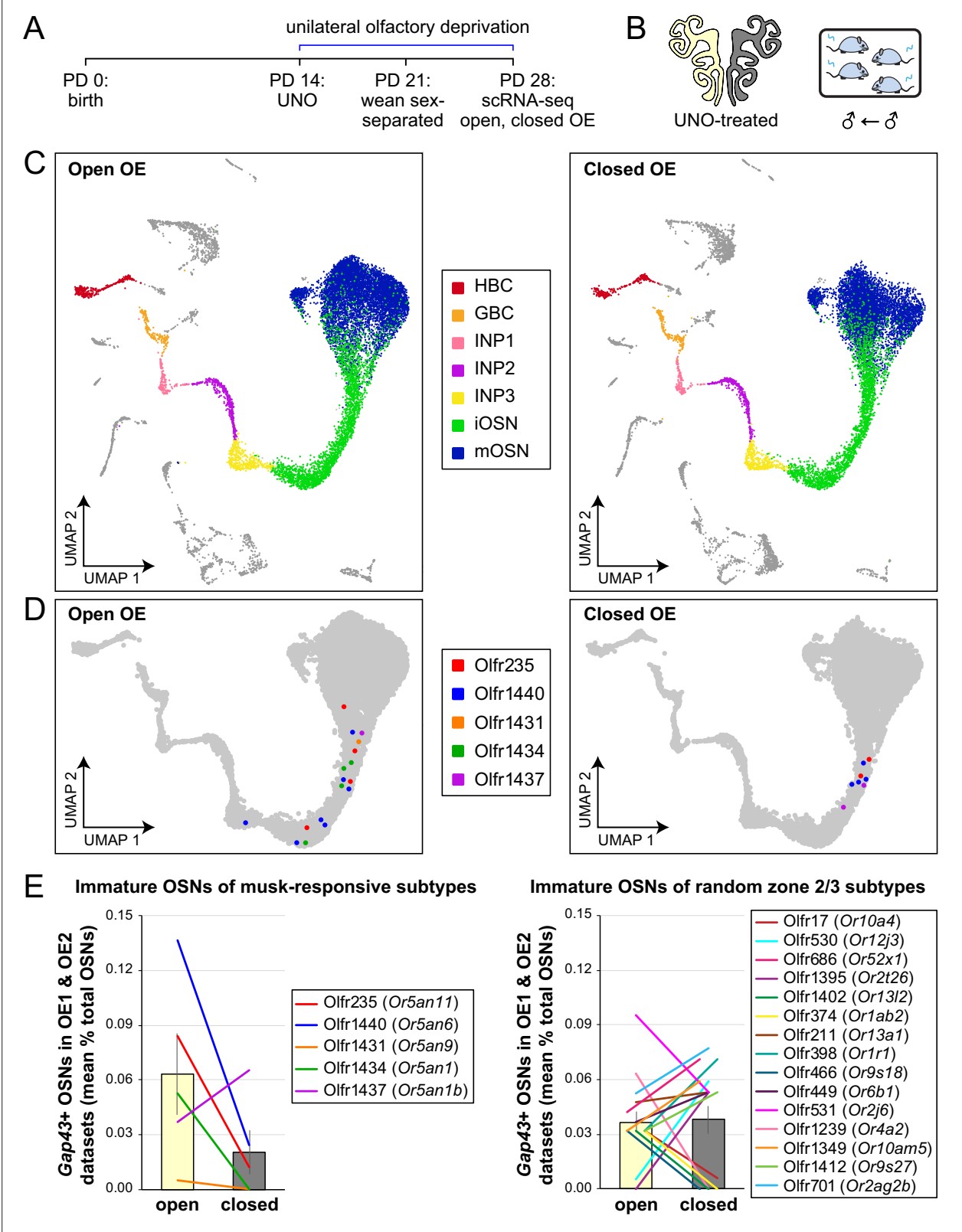

**Figure 1.** scRNA-seq analyses of the open and closed sides of whole OEs from UNO-treated adolescent male mice reveal greater quantities of newborn OSNs of musk-responsive subtypes on the open side of the OE relative to the closed. (**A**, **B**). Experimental timeline (**A**) and conditions (**B**) used to generate scRNA-seq datasets for assessing the effects of UNO on quantities of newborn OSNs of specific subtypes in the OEs of adolescent male mice. Datasets were generated from the open and closed sides of OEs from mice that were UNO-treated at PD 14, weaned sex-separated at PD 21,

*Figure 1 continued on next page*

*Figure 1 continued*

and sacrificed at PD 28 (*van der Linden et al., 2020*). (**C**) A UMAP representation of all cells within the open (*left*) and closed (*right*) side datasets corresponding to OE sample 2 (OE 2). Cells within the OSN lineage are represented by colored dots, as defined in the legend: horizontal basal cells (HBC; *red*), globose basal cells (GBC; *orange*), immediate neuronal precursor 1 cells (INP1; *pink*), immediate neuronal precursor 2 cells (INP2; *purple*), immediate neuronal precursor 3 cells (INP3; *yellow*), immature OSNs (iOSN; *green*), and mature OSNs (mOSN; *blue*). (**D**) UMAP representation of cells within the OSN lineage of the open (*left*) and closed (*right*) side datasets of OE 2. Immature (*Gap43*+) OSNs of the five known musk-responsive subtypes are represented by colored dots, as indicated in the legend. (**E**) Quantification of individual (*lines*) and mean (*bars*) percentages of the OSN population represented by immature OSNs of musk-responsive subtypes (*left*) or randomly chosen zone 2/3 subtypes (*right*) within the open and closed side datasets. Quantifications represent the averages of subtype-specific iOSN quantities obtained from scRNA-seq datasets corresponding to OEs from two different mice (OE 1 and OE 2). Error bars: SEM.

The online version of this article includes the following source data and figure supplement(s) for figure 1:

**Source data 1.** Data associated with *Figure 1E*.

**Figure supplement 1.** scRNA-seq analyses of the open and closed sides of whole OEs from UNO-treated adolescent male mice reveal greater quantities of newborn OSNs of musk-responsive subtypes on the open side of the OE relative to the closed.

**Figure supplement 1—source data 1.** Data associated with *Figure 1—figure supplement 1D, E*.

**Figure supplement 2.** Individual OSNs of musk-responsive subtypes on the open and closed sides of OEs from UNO-treated male mice exhibit typical OR expression within cells of the OSN lineage.

**Figure supplement 2—source data 1.** Data associated with *Figure 1—figure supplement 2*.

*1C–E*). By contrast, newborn OSNs of 15 randomly chosen subtypes located within approximately the same region of the OE where musk-responsive subtypes reside (canonical zones 2 and 3; *Tan and Xie, 2018*) comprise, on average, nearly equal quantities within the open and closed sides (1.07-fold difference; *Figure 1E-right*). To verify that the differences in newborn OSN quantities are not due to aberrant OR expression regulation (e.g. co-expression of multiple ORs), we analyzed cell-specific OR expression at three stages of OSN differentiation: immediate neuronal precursor 3 (INP3), immature OSN (iOSN), and mature OSN (mOSN; *Figure 1—figure supplement 2*). As expected, individual iOSNs and mOSNs of specific subtypes exhibit robust and singular OR expression on both the open and closed sides of OEs from UNO-treated mice, while INP3 cells co-express low levels of multiple OR transcripts, as observed previously (*Fletcher et al., 2017*; *Hanchate et al., 2015*; *Tan et al., 2015*; *Bashkirova et al., 2023*; *Pourmorady et al., 2024*; *Saraiva et al., 2015*; *Scholz et al., 2016*).

To further test whether olfactory deprivation reduces quantities of newborn OSNs of musk-responsive subtypes in mice exposed to male odors, we employed an established histological assay in which EdU-birthdated OSNs of specific subtypes are identified *via* EdU staining and OR-specific RNA-FISH (*van der Linden et al., 2020*; *Hossain et al., 2023*). Using this approach, we quantified the number of newborn OSNs of four musk-responsive subtypes per half-section, on the open and closed sides of OEs from male mice that had been UNO-treated at PD 14, weaned sex-separated at PD 21, EdU-injected at PD 28, and dissected at PD 35 (*Figure 2A and B*). The EdU injection time-point was based on the timepoint used for identification of immature (*Gap43*-expressing) OSNs in the scRNA-seq datasets, while the dissection timepoint was chosen to provide a chase period of 7 days, which is sufficient to enable robust and stable expression of an OR (*van der Linden et al., 2020*). In agreement with findings from scRNA-seq, EdU-labeled OSNs of subtypes Olfr235, Olfr1440, and Olfr1431 exhibited significantly greater quantities on the open side of the OE compared to the closed side in adolescent male mice (p<0.05; 2.3-fold, 1.8-fold, and 2.5-fold, respectively) (*Figure 2C–F*). By contrast, subtype Olfr1437 did not show a significant open-side bias (p>0.05; 1.01-fold; *Figure 2G*), nor did two non-musk-responsive subtypes, Olfr912 and Olfr1463 (*Or5b109*) (1.05-fold for both subtypes; *Figure 2H*; *Figure 2—figure supplement 1*). To verify that observed open-side biases in quantities of newborn OSNs are not an artifact of the normalization method used, which was based on the number of OE half-sections quantified, we compared this method to two alternatives: based on the number of EdU+ cells or on the total DAPI+ area. Notably, all three methods yielded open-side biases of similar magnitude, with no significant differences in UNO effect sizes (defined as the $\log_2$ [open/closed] ratio) observed between the three methods (*Figure 2—figure supplement 2*). Taken together, these findings indicate that olfactory deprivation via UNO selectively reduces quantities of newborn OSNs of some musk-responsive subtypes in adolescent male mice, consistent with the possibility that the birthrates of these subtypes are selectively accelerated by exposure to male odors.

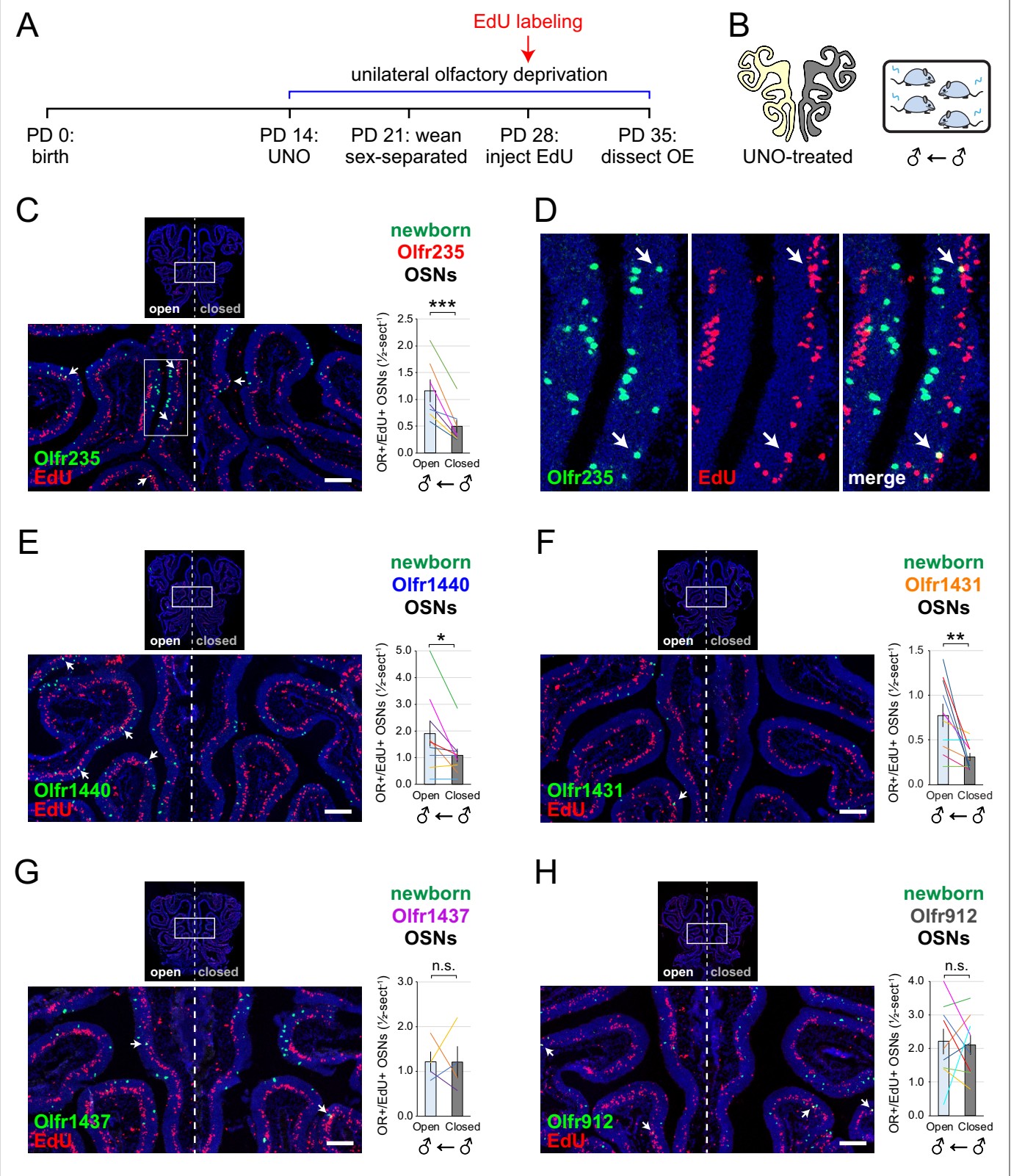

**Figure 2.** EdU birthdating analyses confirm that UNO-treated adolescent male mice exhibit greater quantities of newborn OSNs of specific musk-responsive subtypes on the open side of the OE relative to the closed. (**A, B**) Experimental timeline (**A**) and conditions (**B**) used to generate OE tissue samples for assessing the effects of UNO on quantities of newborn OSNs of specific subtypes in adolescent male mice. Mice were UNO-treated at PD 14, weaned sex-separated at PD 21, EdU-labeled at PD 28, and sacrificed at PD 35. OEs were sectioned and analyzed using OR-specific RNA-FISH

*Figure 2 continued on next page*

*Figure 2 continued*

and EdU staining. (**C**, **E–H**) *Left*: Representative images of OE sections from UNO-treated adolescent male mice that were exposed, at the time of EdU labeling, to male littermates (♂ ← ♂), with newborn OSNs (OR+/EdU+) indicated by white arrows. *Right*: Quantifications of newborn OSNs on the open and closed sides of tissue sections spanning the anterior-posterior lengths of OEs from UNO-treated male mice reveal significant open-side biases in quantities of newborn OSNs of musk-responsive subtypes Olfr235 (**C**), Olfr1440 (**E**), and Olfr1431 (**F**), but not the musk-responsive subtype Olfr1437 (**G**) or the SBT-responsive subtype Olfr912 (**H**). (**D**) Enlarged, split-channel view of boxed region in (**C**), with Olfr235+/EdU+ OSNs indicated by white arrows. Scale bars: 150 μm. Each line represents a distinct mouse (n=4–10 mice [≥5 sections/mouse] per OSN subtype). ***p<0.001; **p<0.01; *p<0.05; n.s. p>0.05; ratio paired two-tailed t-test. Error bars: SEM.

The online version of this article includes the following source data and figure supplement(s) for figure 2:

**Source data 1.** Data associated with *Figure 2*.

**Figure supplement 1.** EdU birthdating analyses show that UNO-treated adolescent male mice do not exhibit significantly different quantities of newborn OSNs of control subtype Olfr1463 on the open side of the OE relative to the closed.

**Figure supplement 1—source data 1.** Data associated with *Figure 2—figure supplement 1*.

**Figure supplement 2.** Comparison of normalization methods for assessing the effects of exposure to male odors on quantities of newborn Olfr235 OSNs within the OEs of UNO-treated male mice.

**Figure supplement 2—source data 1.** Data associated with *Figure 2—figure supplement 2*.

## The exposure of mice to male-specific odors elevates quantities of newborn Olfr235 OSNs

Observations that naris occlusion reduces quantities of newborn OSNs of subtypes Olfr235, Olfr1440, and Olfr1431 in adolescent male mice, together with previous findings that mice exposed to males from weaning until adulthood exhibit elevated representations of all three subtypes, are consistent with the hypothesis that exposure to male odors accelerates the birthrates of these subtypes. If so, we predicted that UNO-induced open-side biases in newborn OSNs of these subtypes may be attenuated in mice that are not exposed to male odors. To test this, we quantified newborn OSNs of subtype Olfr235 in UNO-treated female mice that were housed either sex-separated or sex-combined starting from the age of weaning (PD 21) and thus either unexposed or exposed, respectively, to male odors at the time of EdU labeling (PD 28; *Figure 3A and B*; *Figure 3—figure supplement 1A*). Unlike sex-separated males, sex-separated females showed no significant difference in quantities of newborn Olfr235 OSNs on the open side of the OE compared to the closed (1.1-fold; *Figure 3C and E-left*) and, correspondingly, a significantly lower UNO effect size (p<0.05; 14-fold; *Figure 3E-right*). By contrast, sex-combined females showed a significantly greater quantity of newborn Olfr235 OSNs on the open side of the OE compared to the closed (p<0.01; 2.2-fold) and, consequently, a UNO effect size significantly greater than that observed in sex-separated females (p<0.05; 11-fold), but not significantly different from sex-separated males (0.8-fold; *Figure 3D and E*). Analogous open-side biases and differences in UNO effect sizes were observed for total Olfr235 OSN quantities (*Figure 3—figure supplement 1B*; *Figure 3F*).

Findings that exposure to male odors is required for occlusion-induced reductions in quantities of newborn Olfr235 OSNs are consistent with the hypothesis that a component of male odors can accelerate the birthrate of this subtype. If so, the exposure of non-occluded mice to male odors should increase observed quantities of newborn Olfr235 OSNs. To test this, we compared quantities of newborn Olfr235 OSNs in non-occluded sex-separated females, sex-separated males, and sex-combined females (*Figure 3A and B*). Remarkably, compared to sex-separated females, sex-separated males and sex-combined females exhibited significantly greater quantities of newborn Olfr235 OSNs (p<0.01; 2.2-fold and 3.1-fold, respectively; *Figure 3G*). By contrast, quantities of newborn OSNs of the SBT-responsive subtype Olfr912 showed no significant differences between the three experimental groups (*Figure 3H*). These findings strongly support the conclusion that the exposure of mice to male odors increases quantities of newborn Olfr235 OSNs within the OE.

Unlike subtype Olfr235, subtypes Olfr1440 and Olfr1431 exhibited significant open-side biases in both newborn and total OSN quantities within the OEs of UNO-treated sex-separated males, sex-separated females, and sex-combined females (p<0.05), with no significant differences in UNO effect size observed between the three groups (*Figure 3—figure supplement 1C–F*). As expected, the non-musk-responsive OSN subtypes Olfr912 and Olfr1463 exhibited no significant open-side biases in newborn OSN quantities in any experimental group, although Olfr912 showed significant closed-side

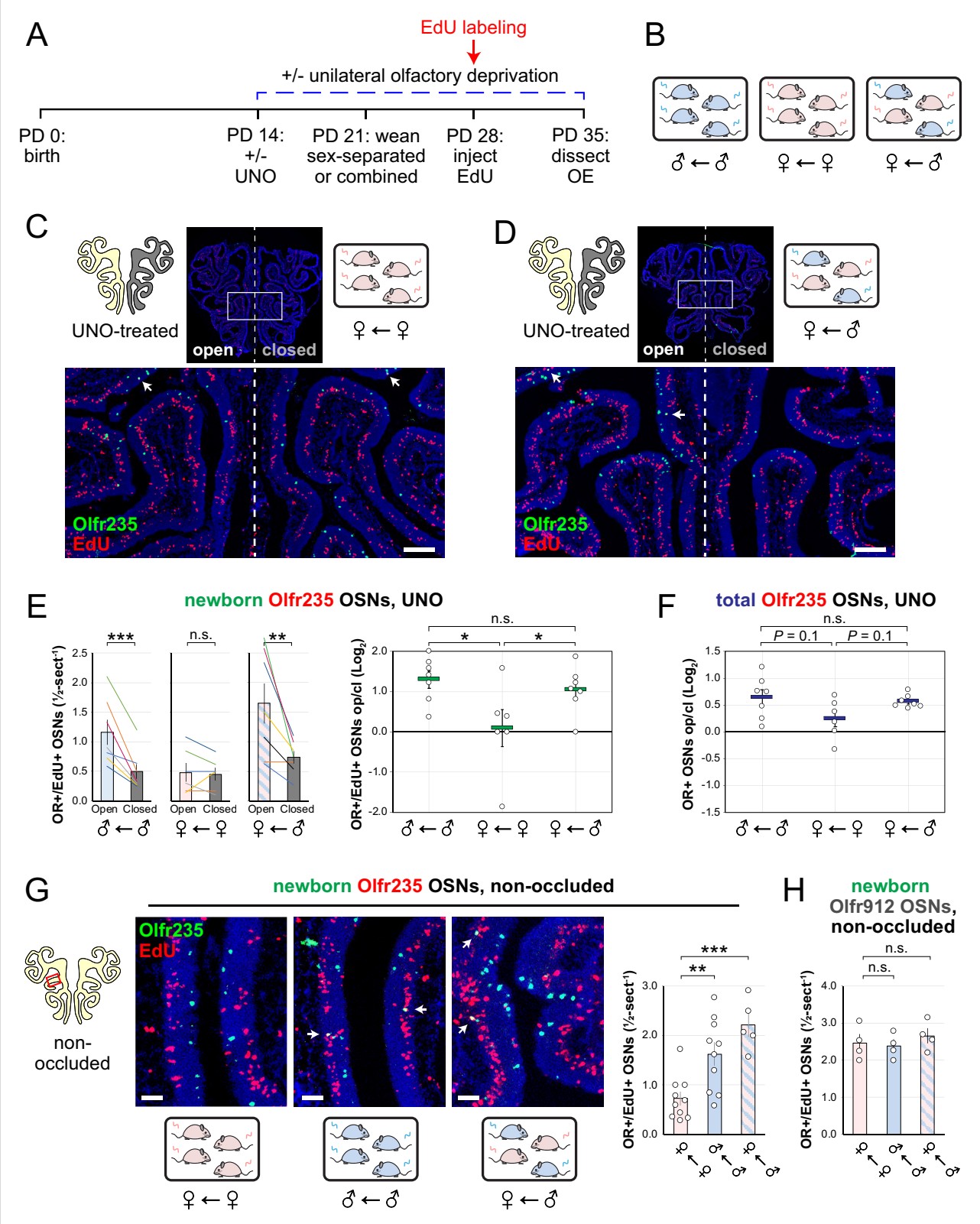

**Figure 3.** Exposure of mice to male mouse odors causes selective increases in quantities of newborn OSNs of the musk-responsive subtype Olfr235. (**A**, **B**) Experimental timeline (**A**) and conditions (**B**) used to generate OE tissue samples for assessing the effects of exposure to male mouse odors on quantities of newborn OSNs of specific subtypes. Male and female mice were either UNO-treated or untreated (non-occluded) at PD 14, weaned sex-separated or sex-combined at PD 21, EdU-labeled at PD 28, and sacrificed at PD 35. OEs were sectioned and analyzed using OR-specific RNA-FISH

*Figure 3 continued on next page*

*Figure 3 continued*

and EdU staining. (**C**, **D**) Representative images of OE sections from UNO-treated female mice that were exposed, at the time of EdU labeling, to either female (♀ ← ♀) (**C**) or male (♀ ← ♂) (**D**) littermates, with newborn Olfr235 OSNs (OR+/EdU+) indicated by white arrows. (**E**, **F**) Quantifications of newborn (**E**) and total (**F**) Olfr235 OSNs on the open and closed sides of tissue sections spanning the anterior-posterior lengths of OEs from UNO-treated mice reveal significant open-side biases in quantities of newborn Olfr235 OSNs in male or female mice exposed to male littermates (♂ ← ♂ or ♀ ← ♂), but not in females exposed to only female littermates (♀ ← ♀) (**E**-*left*), and greater UNO effect sizes for quantities of newborn (**E**-*right*) and total (**F**) Olfr235 OSNs within the OEs of mice that were exposed to male littermates. (**G**-*left, middle*) Representative images (*middle*) corresponding to the region outlined in the schematic (*left, red box*) of OE sections from non-occluded mice that were exposed to male littermates (♂ ← ♂, ♀ ← ♂) or to female littermates (♀ ← ♀) at the time of EdU labeling, with Olfr235+/EdU+ OSNs indicated by white arrows. (**G**-*right*, **H**) Quantifications of newborn Olfr235 (**G**-*right*) and Olfr912 (**H**) OSNs in tissue sections spanning the anterior-posterior lengths of OEs reveal significantly greater quantities of newborn Olfr235 OSNs in non-occluded mice exposed, at the time of EdU labeling, to male littermates (♂ ← ♂, ♀ ← ♂) compared to those exposed only to female littermates (♀ ← ♀), while quantities of newborn Olfr912 OSNs showed no significant differences between mice exposed to male littermates and those exposed only to females. Scale bars: 150 μm (**C**, **D**), 50 μm (**G**). Each line or circle represents a distinct mouse (n=4–10 mice [≥5 sections/mouse] per OSN subtype and condition). ***p<0.001; **p<0.01; *p<0.05; n.s. p>0.05; ratio paired two-tailed t-test (**E**-*left*); one-way ANOVA test, FDR-adjusted (**E**-*right*, **F–H**). Error bars: SEM. Data in (**E**) for newborn OSN quantities in ♂ ← ♂ OE samples correspond to *Figure 2—source data 1*; *Figure 3—source data 1*.

The online version of this article includes the following source data and figure supplement(s) for figure 3:

**Source data 1.** Data associated with *Figure 3*.

**Figure supplement 1.** Effects of exposure to male odors on differences in quantities of newborn and total OSNs of musk-responsive and control subtypes on the open and closed sides of OEs from UNO-treated mice.

**Figure supplement 1—source data 1.** Data associated with *Figure 3—figure supplement 1*.

**Figure supplement 2.** Comparison of normalization methods for assessing the effects of exposure to male odors on quantities of newborn Olfr235 OSNs within the OEs of non-occluded mice.

**Figure supplement 2—source data 1.** Data associated with *Figure 3—figure supplement 2*.

**Figure supplement 3.** UNO-induced open-side biases in quantities of newborn Olfr1431 OSNs are increased by exposure to adult mice.

**Figure supplement 3—source data 1.** Data associated with *Figure 3—figure supplement 3*.

biases in total OSNs (p<0.01; *Figure 3—figure supplement 1G–J*), consistent with the possibility that naris occlusion protects OSNs of this subtype from overstimulation (*van der Linden et al., 2018*; *Vihani et al., 2020*). Observations that exposure to male odors is required for occlusion-induced reductions in quantities of newborn OSNs of subtypes Olfr235 but not Olfr1440 or Olfr1431 were unexpected considering that ORs of all three subtypes show male-biased expression in mice housed sex-separated until 6 months of age (*Appendix 1—figure 1*; *van der Linden et al., 2018*). A conceivable explanation for these findings is that, unlike subtype Olfr235, quantities of newborn Olfr1440 and Olfr1431 OSNs are elevated by exposure to odors emitted by both male and female mice at the adolescent stage. Such differences could reflect variations in the specific odorants to which distinct musk-responsive subtypes are most sensitive (*Sato-Akuhara et al., 2016*), and which may depend on the age and sex of mice contributing to the odor environment (*Osada et al., 2008*; *Osada et al., 2003*; *Schwende et al., 1986*; *Stopková et al., 2023*). If so, we predicted that the age and/or sex of mice within the odor environment at the time of EdU labeling might differentially affect the degree to which quantities of newborn OSNs of these subtypes are affected by olfactory deprivation. To test this, we compared open-side biases in newborn OSN quantities within UNO-treated adolescent males exposed to littermates alone versus those exposed to both littermates and adult parents (*Figure 3—figure supplement 3A, B*). Remarkably, newborn Olfr1431 OSNs exhibited a significantly greater (p<0.05; 2-fold) UNO effect size in the presence of adult mice compared to the absence, reflecting open-side biases of 4.8-fold and 2.5-fold, respectively (*Figure 3—figure supplement 3C, D*). By contrast, UNO effect sizes for newborn OSNs of subtypes Olfr235, Olfr1440, and Olfr1437, as well as the control subtype Olfr912, were not significantly affected by exposure to adults (*Figure 3—figure supplement 3E–I*). These findings are consistent with the possibility that stimulation-dependent changes in the quantities of newborn OSNs of musk-responsive subtypes may vary depending on the age of odor-emitting mice within the environment.

## The exposure of UNO-treated mice to low concentrations of a musk odorant increases open-side biases in quantities of newborn OSNs of musk-responsive subtypes

Evidence that quantities of newborn OSNs of musk-responsive subtypes can be affected by exposure to unknown components of mouse odors indicated that the birthrates of specific OSN subtypes may be accelerated by discrete odorants. If so, we predicted that the exposure of mice to musk odorants that are known to stimulate these subtypes would increase their neurogenesis rates. However, because high levels of chronic stimulation of OSNs have been found to reduce the representations of stimulated subtypes within the OSN population, presumably *via* reductions in OSN lifespan (*Cadiou et al., 2014*; *Cavallin et al., 2010*; *van der Linden et al., 2018*), we speculated that there may be a range of odorant concentrations sufficient to accelerate the birthrates of these subtypes without shortening their OSN lifespans. To assess this, we analyzed the effects of exposing UNO-treated female mice to varying concentrations of the musk odorant (*R*)–3-methylcyclopentadecanone (muscone) on quantities of newborn and total OSNs of the musk-responsive subtypes Olfr235, Olfr1440, and Olfr1431, as well as the control subtypes Olfr912 and Olfr1463, via a metal tea-ball from weaning (PD 21) until dissection (PD 35; *Figure 4A and B*). Remarkably, exposure of mice to muscone was found to significantly affect the extent to which UNO treatment alters the relative quantities of both newborn and total OSNs of musk-responsive subtypes on the open and closed sides of the OE in a concentration-dependent manner (*Figure 4C*; *Figure 4—figure supplements 1 and 2*). Subtype Olfr235, for example, which showed no significant difference in newborn OSN quantities between the open and closed sides in unexposed females, exhibited significant open-side biases in females exposed to 0.1% or 1% muscone ($p<0.01$; 2.4 and 2.0-fold, respectively), but not those exposed to 10% (1.2-fold; *Figure 4C-middle*). Accordingly, mice exposed to 0.1% or 1% muscone exhibited significantly greater UNO effect sizes for quantities of newborn Olfr235 OSNs compared to unexposed females ($p<0.05$; 14 and 13-fold, respectively; *Figure 4C-right*). Similarly, open-side biases in total Olfr235 OSNs were observed in mice exposed to 0.1% or 1% muscone, with the most significant effect observed at 0.1% ($p<0.01$; *Figure 4—figure supplement 2*). One possible explanation for the reduction in UNO effect sizes observed at 10% muscone relative to lower concentrations is that small amounts of odorant may enter the closed side of the OE due to transnasal and/or retronasal odor transfer via the nasopharyngeal canal (*Kelemen, 1947*) and naso-pharynx, respectively (*Coppola, 2012*), an effect that would be expected to increase with environmental concentrations and attenuate differences in observed effects of odor stimulation between the open and closed sides.

Compared to subtype Olfr235, open-side biases in quantities of newborn and total OSNs of subtypes Olfr1440 and Olfr1431 were more subtly affected by the concentration of muscone to which mice were exposed (*Figure 4—figure supplements 1 and 2*). We speculate that this may be explained in part by the fact that both subtypes exhibit significant open-side biases in UNO-treated female mice even in the absence of exogenous odor exposure, which may limit the influence of muscone exposure on the UNO effect size. In the case of Olfr1440, the subtype most sensitive to muscone (*Sato-Akuhara et al., 2016*; *Shirasu et al., 2014*), it is conceivable that mature OSNs are robustly stimulated on the closed side of the OE by very small amounts of odors that enter transnasally and/or retronasally. Consistent with this possibility, UNO-treated females exposed to 0.1% muscone show greater quantities of newborn Olfr1440 OSNs on both the open and closed sides of the OE compared to their unexposed counterparts (*Figure 4—figure supplement 1A-middle*). Relatedly, subtype Olfr1440 showed evidence of shortened OSN lifespans in mice exposed to 1% and 10% muscone, as reflected in the higher total OSN quantities observed on the closed side of the OE (*Figure 4—figure supplement 2B*). In the case of Olfr1431, newborn OSNs exhibited a significantly greater open-side bias in mice exposed to 1% muscone compared to unexposed controls ($p<0.01$; 1.6-fold), but no significant difference at 0.1%, perhaps indicating that this subtype is less sensitive to muscone than other subtypes. Like subtype Olfr235, neither Olfr1440 nor Olfr1431 exhibited significant open-side biases in newborn OSN quantities in mice exposed to 10% muscone, potentially reflecting substantial trans-nasal and/or retronasal odor transfer to the closed side of the OE at this concentration. As expected, muscone exposure did not significantly affect open-side biases in newborn or total OSN quantities of the non-musk responsive subtypes Olfr912 or Olfr1463 (*Figure 4—figure supplements 1 and 2*). Taken together, these findings reveal that the exposure of UNO-treated mice to low concentrations

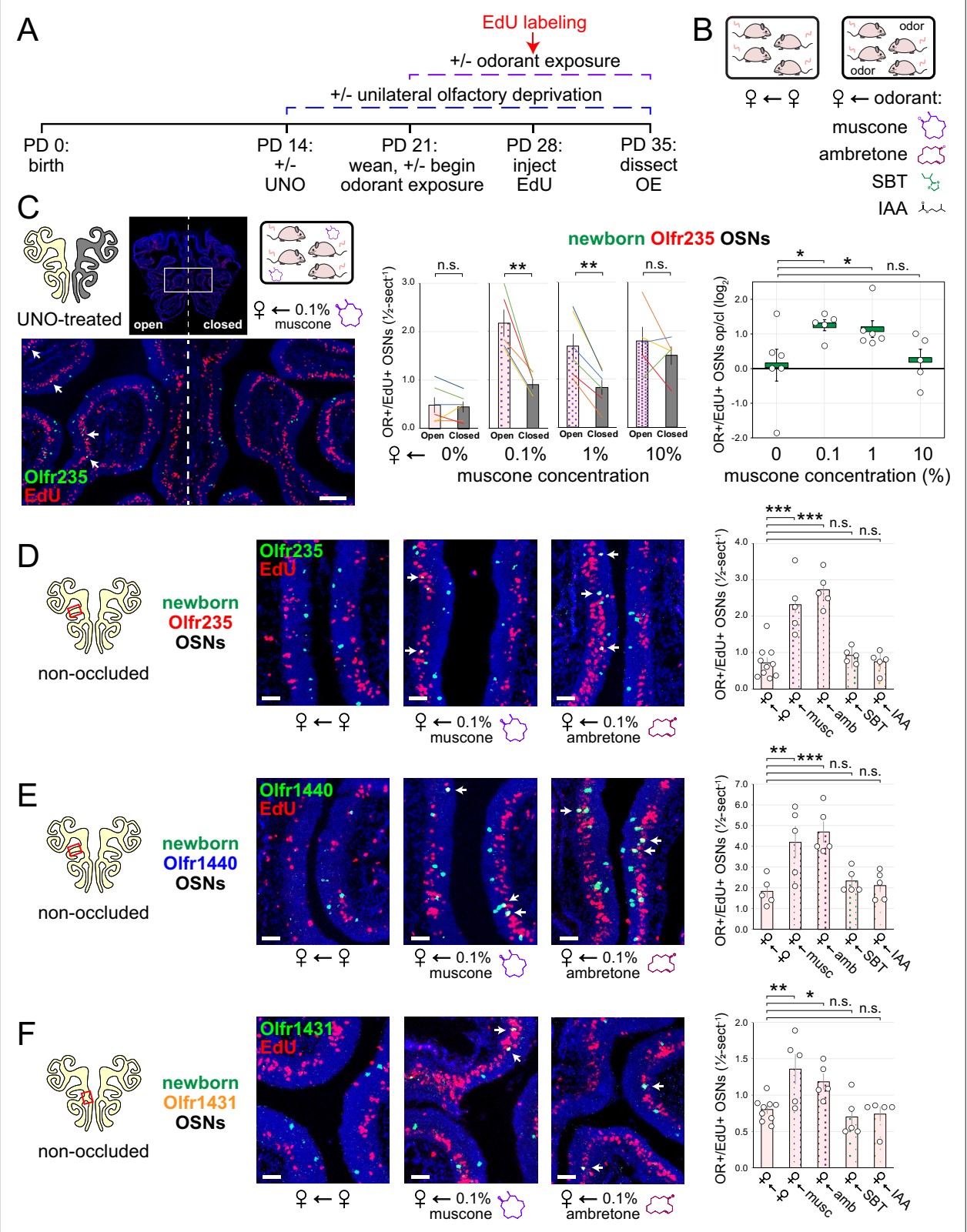

**Figure 4.** Exposure of mice to musk odors causes selective increases in quantities of newborn OSNs of musk-responsive subtypes. (**A**, **B**) Experimental timeline (**A**) and conditions (**B**) used to generate OE tissue samples for assessing the effects of exposure to musk (muscone, ambretone) or non-musk (SBT, IAA) odorants on quantities of newborn OSNs of specific subtypes. Female mice were either UNO-treated or untreated (non-occluded) at PD 14, either exposed or unexposed to an exogenous musk or non-musk odorant starting at PD 21, EdU-labeled at PD 28, and sacrificed at PD 35. OEs were

*Figure 4 continued on next page*

*Figure 4 continued*

sectioned and analyzed using OR-specific RNA-FISH and EdU staining. (**C**) *Left*: Representative image of an OE section from a UNO-treated female mouse that was exposed, at the time of EdU labeling, to 0.1% muscone, with newborn Olfr235 OSNs (OR+/EdU+) indicated by white arrows. *Middle, right*: Quantifications of newborn Olfr235 OSNs on the open and closed sides of tissue sections spanning the anterior-posterior lengths of OEs from UNO-treated female mice reveal significant open-side biases (*middle*) and significantly greater UNO effect sizes (*right*) in mice exposed to 0.1% or 1% (but not 10%) muscone compared to 0%, with a maximum effect size observed at a concentration of 0.1%. (**D–F**) *Left, middle*: Representative images (*middle*) corresponding to the regions outlined in schematics (*left, red box*) of OE sections from non-occluded female mice that were exposed just to female littermates (♀ ← ♀) or also to 0.1% muscone or 0.1% ambretone, with newborn Olfr235 (**D**), Olfr1440 (**E**), and Olfr1431 (**F**) OSNs (OR+/EdU+) indicated by white arrows. *Right*: Quantifications of newborn Olfr235 (**D**), Olfr1440 (**E**), and Olfr1431 (**F**) OSNs in tissue sections spanning the anterior-posterior lengths of OEs reveal that, compared to females exposed just to female littermates (♀ ← ♀), significantly greater quantities of newborn OSNs of all three subtypes were observed in females also exposed to muscone (musc; 0.1%) or ambretone (amb; 0.1%), but not the non-musk odorants SBT (0.1%) or IAA (0.1%). Scale bars: 150 µm (**C**), 50 µm (**D–F**). Each line or circle represents a distinct mouse (n=5–10 mice [≥5 OE sections/mouse] per OSN subtype and condition). ***p<0.001; **p<0.01; *p<0.05; n.s. p>0.05; ratio paired two-tailed t-test (**C**-*middle*); one-way ANOVA test, FDR-adjusted (**C**-*right*, **D–F**). Error bars: SEM. Data for ♀ ← 0% muscone samples in (**C**) correspond to *Figure 3E*. Image and data for ♀ ← ♀ samples (**D**) correspond to *Figure 3G*.

The online version of this article includes the following source data and figure supplement(s) for figure 4:

**Source data 1.** Data associated with *Figure 4*.

**Figure supplement 1.** Exposure of UNO-treated female mice to muscone causes concentration-dependent differences in quantities of newborn OSNs of musk-responsive subtypes on the open and closed sides of the OE.

**Figure supplement 1—source data 1.** Data associated with *Figure 4—figure supplement 1*.

**Figure supplement 2.** Muscone exposure induces concentration-dependent differences in quantities of total OSNs of musk-responsive subtypes on the open and closed sides of OEs from UNO-treated mice.

**Figure supplement 2—source data 1.** Data associated with *Figure 4—figure supplement 2*.

**Figure supplement 3.** Comparison of normalization methods for assessing the effects of exposure to a musk odor on subtype-specific newborn OSN quantities within the OEs of non-occluded mice.

**Figure supplement 3—source data 1.** Data associated with *Figure 4—figure supplement 3*.

**Figure supplement 4.** Effects of exposure of mice to musk and non-musk odors on quantities of newborn OSNs of non-musk-responsive subtypes, including those previously found to undergo stimulation-dependent changes in OSN birthrate.

**Figure supplement 4—source data 1.** Data associated with *Figure 4—figure supplement 4*.

of a musk odorant is sufficient to induce open-side biases in the quantities of newborn OSNs of musk-responsive subtypes without apparent adverse effects on OSN survival.

## The exposure of non-occluded mice to musk odorants selectively elevates quantities of newborn OSNs of musk-responsive subtypes

Findings that the exposure of UNO-treated mice to muscone concentrations as low as 0.1% can increase open-side biases in quantities of newborn OSNs of musk-responsive subtypes suggested that the exposure of non-occluded mice to musk odorants at this concentration might selectively accelerate the birthrates of these subtypes. To test this, we compared quantities of newborn OSNs of three musk-responsive subtypes and two musk non-responsive subtypes within the OEs of non-occluded female mice exposed to a 0.1% concentration of one of two different musk odorants, muscone or 5-cyclohexadecenone (ambretone), one of two different non-musk odorants, SBT or isoamyl acetate (IAA), or no odorant. Relative to unexposed controls, all three musk-responsive subtypes showed significantly elevated quantities of newborn OSNs in mice exposed to muscone and ambretone (p<0.05), with Olfr235 exhibiting increases of 3.2 and 3.8-fold, Olfr1440 exhibiting increases of 2.3 and 2.6-fold, and Olfr1431 exhibiting increases of 1.7 and 1.5-fold, respectively (*Figure 4D–F*). Notably, these differences were not affected by the use of alternative normalization methods (*Figure 4—figure supplement 3*). By contrast, none of the three musk-responsive subtypes exhibited significant changes in newborn OSN quantities in mice exposed to the non-musk odorants SBT or IAA. Likewise, the SBT-responsive subtype Olfr912 exhibited no significant changes in quantities of newborn OSNs in mice exposed to muscone, ambretone, or SBT (*Figure 4—figure supplement 4C-left*). Moreover, two non-musk-responsive subtypes that were previously found to undergo occlusion-dependent reductions in neurogenesis (*van der Linden et al., 2020*), Olfr827 (*Or9k7*) and Olfr1325 (*Or13ae2*), showed no significant changes in quantities of newborn OSNs in mice exposed to muscone (*Figure 4—figure*

*supplement 4C-middle, right*). Taken together, these findings indicate that: (1) the exposure of non-occluded mice to exogenous musk odorants selectively increases quantities of newborn OSNs of musk-responsive subtypes, (2) the exposure of mice to exogenous non-musk odorants has no effect on quantities of newborn OSNs of musk-responsive subtypes, and (3) only a fraction of subtypes have a capacity to exhibit increases in quantities of newborn OSNs following exposure to their cognate odorants.

## Musk odorant-dependent increases in quantities of newborn OSNs of musk-responsive subtypes persist into adulthood

Findings that the exposure of adolescent mice to male or musk-like odors elevates quantities of newborn OSNs of musk-responsive subtypes raise the question of whether this phenomenon is limited to early life or, rather, persists into adulthood. To address this, we compared quantities of newborn OSNs of three musk-responsive subtypes and one musk non-responsive subtype within the OEs of 9-week-old (PD 65) non-occluded female mice that had been either unexposed to an exogenous odorant or exposed to 0.1% muscone starting from weaning (PD 21) and EdU-treated for 3 days starting at 8 weeks of age (PD 56–58; *Figure 5A and B*). Remarkably, despite the generally reduced OSN birthrate in adult mice compared to adolescents, all three musk-responsive subtypes showed elevated quantities of newborn OSNs in mice exposed to muscone relative to unexposed controls, with Olfr235, Olfr1440, and Olfr1431 exhibiting 1.5-fold, 2.2-fold, and 1.9-fold increases, respectively (*Figure 5C–E*). Notably, observed increases reached statistical significance ($p<0.05$) only for subtype Olfr1440, likely due to the higher baseline birthrate of this subtype relative to Olfr235 and Olfr1431. As expected, exposure to muscone did not significantly increase quantities of newborn OSNs of subtype Olfr912 (*Figure 5F*). These findings indicate that the capacity for odorant-dependent increases in quantities of newborn OSNs of specific subtypes persists into adulthood.

## The time-independence of odor-driven increases in EdU-labeled OSNs of musk-responsive subtypes is consistent with a mechanism involving altered OSN birthrates

Odor-dependent increases in the quantities of newborn OSNs of musk-responsive subtypes could, in principle, be caused by a mechanism that selectively accelerates the rates with which these subtypes are generated or, alternatively, the rates with which they are selectively enriched following their generation (e.g. via enhanced survival or OR switching *Shykind et al., 2004*). If differences in newborn OSN quantities are mediated by selective enrichment, they would be expected to increase in magnitude over time following EdU labeling as specific newborn OSNs exhibit longer lifespans or switch their OR identities in the presence of stimulation. If, however, changes are mediated by accelerated birthrates of specific OSN subtypes, increases in newborn OSN quantities should appear shortly following EdU labeling and remain stable over time. To distinguish between these possibilities, we compared stimulation-dependent changes in quantities of newborn musk-responsive OSNs at two timepoints: 4 days post-EdU, the earliest point during OSN differentiation when OR transcripts can be consistently detected *via* RNA-FISH (*van der Linden et al., 2020*; *Rodriguez-Gil et al., 2015*), and three days later (7 days post-EdU; *Figure 6A*). In initial experiments, the time dependence of open-side biases in quantities of EdU-labeled OSNs of musk-responsive subtypes was assessed in UNO-treated and sex-separated males, females, and females exposed to 0.1% muscone (*Figure 6B*). In sex-separated males, robust open-side biases and statistically indistinguishable UNO effect sizes were observed at 4 and 7 days post-EdU for quantities of EdU-labeled OSNs of subtypes Olfr235, Olfr1440, and Olfr1431 (*Figure 6C and D*; *Figure 6—figure supplement 1A, B*). Likewise, UNO-treated and muscone-exposed females exhibited robust open-side biases in quantities of newborn Olfr235 OSNs and statistically indistinguishable UNO effect sizes at 4 and 7 days post-EdU (*Figure 6E*; *Figure 6—figure supplement 1C*). As expected, no significant open-side biases in EdU-labeled OSN quantities or UNO effect sizes were observed at either 4 or 7 days post-EdU for subtypes Olfr912 or Olfr1463 in UNO-treated male (*Figure 6—figure supplement 1D*, E) or female mice exposed to muscone (*Figure 6—figure supplement 1F*), or for subtype Olfr235 in sex-separated females (*Figure 6—figure supplement 1G*).

Using a similar approach, we also assessed the time dependence of differences in muscone-driven increases in quantities of EdU-labeled OSNs of specific subtypes in non-occluded female mice

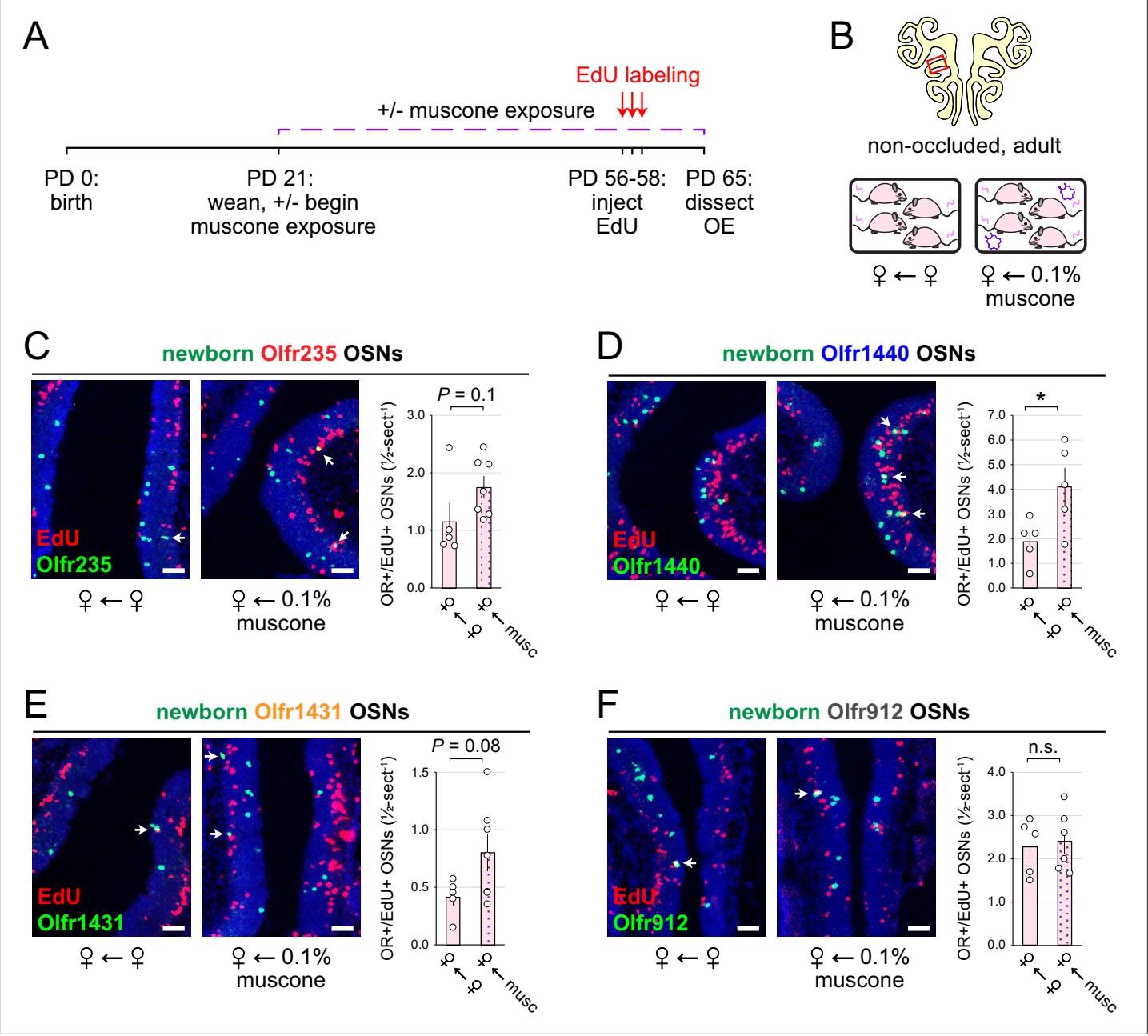

**Figure 5.** Musk exposure increases quantities of newborn OSNs of musk-responsive subtypes in adulthood. (**A**, **B**) Experimental timeline (**A**) and conditions (**B**) used to generate OE tissue samples for assessing the effects of exposure to musk odors on quantities of newborn OSNs of specific subtypes in non-occluded adult mice. Mice were either exposed or unexposed to 0.1% muscone starting at PD 21, EdU-labeled at PD 56–58, and sacrificed at PD 65. OEs were sectioned and analyzed using OR-specific RNA-FISH and EdU staining. (**C–F**) *Left*: Representative images corresponding to the region outlined in the schematic in (**B**-*top*), of OEs from non-occluded adult female mice that were either exposed to just female littermates (♀ ← ♀) or also to muscone (♀ ← 0.1% muscone), with newborn Olfr235 (**C**), Olfr1440 (**D**), Olfr1431 (**E**), and Olfr912 (**F**) OSNs (OR+/EdU+) indicated by white arrows. *Right*: Quantifications of newborn OSNs within tissue sections spanning the anterior-posterior lengths of OE from non-occluded adult female mice reveal greater quantities of newborn OSNs of all three musk-responsive subtypes (Olfr235, Olfr1440, and Olfr1431) in mice exposed to muscone compared to just female littermates (♀ ← ♀), while quantities of newborn OSNs of the SBT-responsive subtype Olfr912 were relatively unaffected. Scale bars: 50 μm. Each circle represents a distinct mouse (n=5–7 mice [≥5 OE sections/mouse] per OSN subtype and condition). *p<0.05; n.s. p>0.05; unpaired two-tailed *t*-test. Error bars: SEM.

The online version of this article includes the following source data for figure 5:

**Source data 1.** Data associated with *Figure 5*.

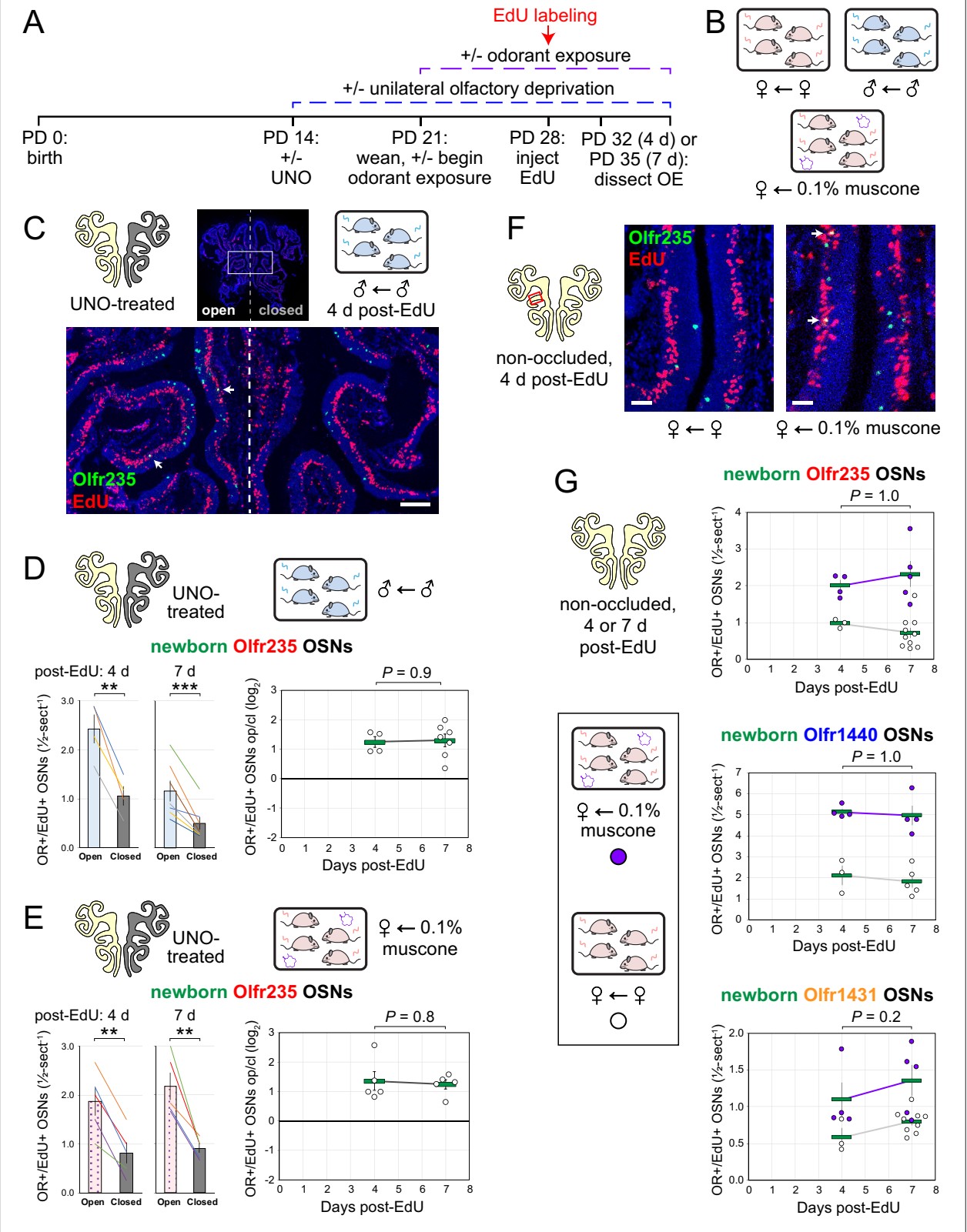

**Figure 6.** Stimulation-dependent increases in quantities of newborn OSNs of musk-responsive subtypes are stable over time following birth, consistent with a mechanism involving altered neurogenesis. (**A**, **B**) Experimental timeline (**A**) and conditions (**B**) used to generate OE tissue samples for assessing the time-dependence of the effects of exposure to male or musk odors on quantities of newborn OSNs of specific subtypes. Mice were either UNO-treated or untreated (non-occluded) at PD 14, either exposed or unexposed to muscone starting at PD 21, EdU-labeled at PD 28, and sacrificed at either

*Figure 6 continued on next page*

*Figure 6 continued*

PD 32 (4 d post-EdU) or PD 35 (7 d post-EdU). OEs were sectioned and analyzed using OR-specific RNA-FISH and EdU staining. (**C**) Representative image of an OE section from a UNO-treated male mouse that was exposed, at the time of EdU labeling, to male littermates (♂ ← ♂) and sacrificed 4 d post-EdU, with newborn Olfr235 OSNs (OR+/EdU+) indicated by white arrows. (**D, E**) Quantifications of newborn Olfr235 OSNs on the open and closed sides of tissue sections spanning the anterior-posterior lengths of OEs from UNO-treated male mice that were exposed to male littermates (♂ ← ♂) (**D**) or female mice that were exposed to 0.1% muscone (♀ ← 0.1% muscone) (**E**) and sacrificed 4 or 7 d post-EdU. Under both conditions, significant open-side biases in quantities of newborn Olfr235 OSNs were observed at both timepoints (*left*), with no statistically significant differences in UNO effect sizes observed over time (*right*). (**F**) Representative images (*right*) corresponding to the region outlined in schematic (*left, red box*) of OE sections from non-occluded female mice that were exposed, at the time of EdU labeling, to either just their female littermates (♀ ← ♀) or also to muscone (♀ ← 0.1% muscone) and sacrificed 4 d post-EdU, with newborn Olfr235 OSNs (OR+/EdU+) indicated by white arrows. (**G**) Quantifications of newborn Olfr235 (*top*), Olfr1440 (*middle*), and Olfr1431 (*bottom*) OSNs within tissue sections spanning the anterior-posterior lengths of OEs from non-occluded female mice that were exposed to either just their female littermates (♀ ← ♀) or also to muscone (♀ ← 0.1% muscone) and sacrificed 4 or 7 d post-EdU. Greater quantities of newborn OSNs of all three subtypes were observed at both timepoints in muscone-exposed mice compared to controls, with no statistically significant interaction between time and newborn OSN quantities observed. Scale bars: 150 μm (C), 50 μm (F). Each circle represents a distinct mouse (n=4–10 mice [≥5 OE sections/mouse] per OSN subtype and condition). ***p<0.001; **p<0.01; ratio paired two-tailed t-test (**D, E**-*left*); unpaired two-tailed t-test (**D, E**-*right*); two-way ANOVA test (factors: muscone exposure, days post-EdU) (**G**) Error bars: SEM. Data for 7 d post-EdU samples correspond to *Figures 2–4*.

The online version of this article includes the following source data and figure supplement(s) for figure 6:

**Source data 1.** Data associated with *Figure 6*.

**Figure supplement 1.** Stimulation-dependent increases in quantities of newborn OSNs of musk-responsive subtypes are stable over time following neurogenesis, consistent with a mechanism involving altered birthrate.

**Figure supplement 1—source data 1.** Data associated with *Figure 6—figure supplement 1*.

(*Figure 6G-left*). Consistent with UNO-based findings, non-occluded mice exhibited statistically indistinguishable muscone-dependent increases at 4 and 7 days post-EdU for quantities of newborn OSNs of the musk-responsive subtypes Olfr235, Olfr1440, and Olfr1431 (*Figure 6G-right*), as well as the control subtype Olfr912 (*Figure 6—figure supplement 1H*). Taken together, these findings provide evidence that odor-dependent increases in the quantities of newborn OSNs of musk-responsive subtypes reflect selective changes in the birthrates of these subtypes.

## Evidence that mice emit musk-like odors

Previous findings that OSNs of musk-responsive subtypes respond to male mouse odors (*van der Linden et al., 2018*) and become more highly represented within the OEs of mice exposed to males (*van der Linden et al., 2018*; *Vihani et al., 2020*), together with evidence from the current study that the exposure of mice to male odors can accelerate the birthrates of these subtypes, indicate that mice emit odors that stimulate musk-responsive subtypes. Interestingly, mouse preputial glands have been found to have a high degree of histomorphological, transcriptomic, and molecular similarities to muskrat scent glands (*Han et al., 2022*), which secrete musk odors (*Mookherjee and Wilson, 2012*). Moreover, a previous study noted preliminary evidence that mouse preputial gland extracts contain compounds that activate Olfr1440 OSNs (*Sato-Akuhara et al., 2016*). Based on this information, we analyzed mouse preputial gland extracts *via* gas chromatography – mass spectrometry (GC-MS) for known musk molecules, particularly those shown to stimulate Olfr235 and Olfr1440 OSNs (*Sato-Akuhara et al., 2016*). Intriguingly, these analyses revealed the presence of molecules whose GC-MS signals are structurally consistent with known musk molecules (*Appendix 1—figure 2*). A potential match to one such signal is cycloheptadecanol, a musk molecule that is structurally similar to those known to activate Olfr235 and Olfr1440 OSNs (*Sato-Akuhara et al., 2016*). Verification of the presence of this and other musk-like odors that correspond to signals from mouse preputial glands will require comparisons to pure standards. Notably, the approach employed here was solely focused on the preputial gland and known musk molecules. However, it is conceivable that mouse musk-responsive OSNs are naturally stimulated by molecules that are structurally unrelated to those that have been characterized to date or are emitted from sources other than preputial glands. Future studies will be needed to address these possibilities.

## Discussion

### The exposure of mice to discrete odorants can selectively accelerate the birthrates of OSN subtypes that they stimulate

Previous studies have established that UNO reduces the overall rate of neurogenesis on the closed side of the OE relative to the open (*Farbman et al., 1988*; *Cummings and Brunjes, 1994*; *Mirich and Brunjes, 2001*; *Suh et al., 2006*). Recently, these occlusion-induced reductions were found to reflect decelerations of the birthrates of only a fraction of the ~1200 OSN subtypes and to vary according to age and/or the olfactory environment (*van der Linden et al., 2020*), suggesting the possibility that unknown olfactory stimuli selectively promote the birthrates of these subtypes. In this study, we aimed to test this by elucidating the nature of stimuli that accelerate the birthrates of specific OSN subtypes. We have presented findings that these stimuli include discrete odorants that selectively activate the same subtypes whose birthrates are accelerated. These findings build upon previous observations that a small group of musk-responsive OSN subtypes are more highly represented in the OEs of mice exposed to male odors compared to mice isolated from them (*van der Linden et al., 2018*; *Vihani et al., 2020*), and that some of these subtypes are also responsive to male-specific odors (*van der Linden et al., 2018*; *Appendix 1—figure 1*). These data suggested that one or more components of mouse-emitted odors might naturally accelerate the birthrates of these OSN subtypes. Here, using scRNA-seq-based and histological approaches, we have found that UNO treatment of adolescent male mice reduces quantities of newborn OSNs of musk-responsive subtypes on the closed side of the OE relative to the open, consistent with the possibility that these subtypes have a capacity to undergo stimulation-dependent neurogenesis (*Figures 1 and 2*). Additionally, we have presented findings that the exposure of mice to male-specific odors or two different exogenous musk odorants can selectively elevate quantities of newborn OSNs of musk-responsive subtypes (*Figures 3–5*). Finally, we have described evidence that odor-driven increases in newborn OSN quantities are stable over time after EdU labeling, indicating that the observed changes reflect altered OSN birthrates as opposed to altered OSN lifespan or OR gene switching (*Figure 6*). Collectively, these findings support the hypothesis that the stimuli that regulate the birthrates of specific subtypes are discrete odors that selectively stimulate those subtypes.

### How do discrete odors accelerate the birthrates of specific OSN subtypes?

Our findings that specific OSN subtypes exhibit accelerated birthrates following the exposure of mice to odors that selectively stimulate OSNs of these subtypes indicate that this process occurs via a mechanism that is highly specific with respect to the stimulating odors and the subtypes whose birthrates are accelerated. Considering that horizontal basal cells (HBCs) and globose basal cells (GBCs), the stem and progenitor cells that give rise to new OSNs, lack ORs and signal transduction molecules needed to detect and respond to odors, we hypothesize the existence of a signaling pathway from mature OSNs to HBCs or GBCs that alters the rates at which OSNs of specific subtypes are born. Findings from the present study and a previous one (*van der Linden et al., 2020*) indicate that this signaling capacity may be limited to only a fraction of OSN subtypes, since a majority of subtypes do not appear to exhibit stimulation-dependent neurogenesis (*van der Linden et al., 2020*). Subtype Olfr912, for example, which detects the male-specific odor component SBT (*Vihani et al., 2020*), exhibits no increase in birthrate upon exposure of female mice to male odors and was therefore employed in this study as a control subtype. We speculate that the receipt of odor-derived signals by HBCs or GBCs alters OR choices or amplifies choices that have already been made. Elucidating the nature of odor-dependent signals received by HBCs/GBCs and the mechanism by which these signals accelerate the birthrates of specific OSN subtypes are important areas of future investigation.

### What functions does discrete odor stimulation-dependent neurogenesis of specific OSN subtypes serve?

Because OSN differentiation entails the stochastic process of singular OR choice (*McClintock, 2015*; *Monahan and Lomvardas, 2015*), it has long been assumed that OSN neurogenesis is entirely stochastic with respect to OR identity. Thus, unlike other regions of the nervous system where persistent neurogenesis is known to play important adaptive roles (*Lledo and Valley, 2016*; *Ming and*

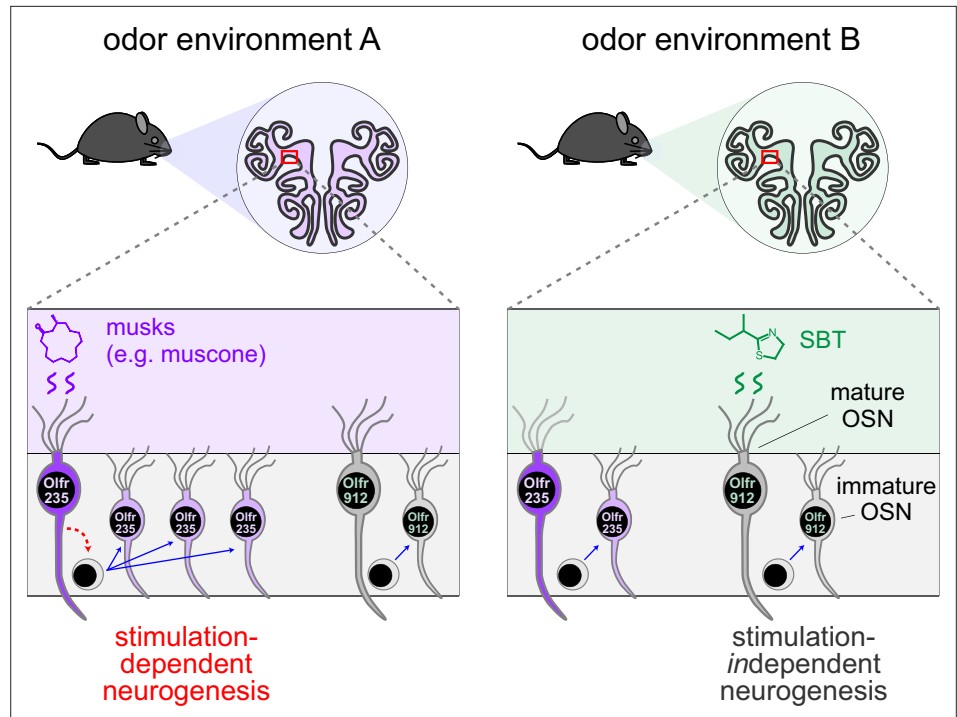

**Figure 7.** Model for how specific odors selectively increase quantities of newborn OSNs of subtypes that are stimulated by them. A fraction of OSN subtypes (e.g. Olfr235), upon stimulation by discrete odors (e.g. musks), undergo accelerated rates of neurogenesis. Most subtypes (e.g. Olfr912) do not exhibit altered rates of neurogenesis upon stimulation by odors that stimulate them (e.g. SBT). One hypothetical mechanism involves selective signaling from odor-stimulated mature OSNs of specific subtypes to neural progenitors.

*Song, 2011*; *Opendak and Gould, 2015*), life-long neurogenesis within the OE is generally assumed to serve the merely homeostatic function of replacing neurons lost to turnover and injury (*Yu and Wu, 2017*). Results of the present study, together with those of a previous one (*van der Linden et al., 2020*), challenge these assumptions by demonstrating that neurogenesis is not entirely stochastic with respect to subtype, but rather that the birthrates of a fraction of OSN subtypes can be selectively and directionally regulated by discrete odor experiences (*Figure 7*). These findings suggest the possibility that persistent neurogenesis within the OE serves an unknown adaptive function in addition to the known homeostatic one. It is conceivable, for example, that the acceleration of the birthrates of specific OSN subtypes could selectively enhance sensitivity to odors detected by those subtypes by increasing their representations within the OE (*van Drongelen et al., 1978*; *Meisami, 1989*; *D'Hulst et al., 2016*; *Apfelbach et al., 1991*). Under this scenario, OSNs of affected subtypes might have baseline representations that lie within the dynamic range for signaling to projection neurons under physiological concentrations of cognate odors, such that accelerated neurogenesis could enhance an animal's sensitivity to odors detected by these subtypes. This effect could have relevance to intriguing and unexplained observations in both rodents and humans that exposure to specific odors can dramatically increase sensitivity to them (*Wang et al., 1993*; *Yee and Wysocki, 2001*; *Dalton et al., 2002*; *Voznessenskaya et al., 1995*; *Wang et al., 2004*; *Wysocki et al., 1989*). Alternatively, or in addition, OSNs produced via odor-dependent neurogenesis could conceivably enable the formation of new OB glomeruli and synaptic connections with projection neurons (*Zou et al., 2009*; *Yamada et al., 2017*; *Dorrego-Rivas and Grubb, 2022*; *Qiu et al., 2021*). Under this scenario, stimulation-dependent neurogenesis of specific subtypes could alter inputs to the olfactory cortex and thereby regulate the perception of, and behavioral responses to, specific odors.

## A special role for musk-responsive OSN subtypes

Results of the present study demonstrate that the birthrates of musk-responsive subtypes can be regulated by exposure to musk odors, a group of molecules that are naturally emitted by numerous

mammalian species (*Asada et al., 2012*; *Ruzicka et al., 1926*; *Ward and van Dorp, 1981*). For some mammals, musk odors are known to function in attracting mates, marking territory, and deterring predators (*Mookherjee and Wilson, 2012*; *Agosta, 1992*). Moreover, exposure to musk odors has been reported to cause physiological changes in some mammals, including humans, suggesting that they can function as semiochemicals (*Fukui et al., 2007*; *Kato et al., 2004*). In mice, the physiological functions of musk odors, if any, are unknown, although they have been found to be selectively attractive to male mice (*Horio et al., 2019*). Previous findings that exogenous musk odors activate a small number of related and evolutionarily conserved mouse ORs (*McClintock et al., 2014*; *Sato-Akuhara et al., 2016*; *Shirasu et al., 2014*) and that odors emitted selectively by male mice activate OSN subtypes that express a subset of these ORs (*van der Linden et al., 2018*) suggest the possibility that mice also emit musk-like molecules. Our preliminary findings that mouse preputial gland extracts contain molecules that are structurally related to known musk odors provide additional support for this possibility.

Our findings that the exposure of mice to odors from adolescent males, but not females, accelerates the birthrate of Olfr235 OSNs provide a mechanistic explanation for previous observations that mice exposed to male odors exhibit higher representations of this subtype (*van der Linden et al., 2018*; *Vihani et al., 2020*). Curiously, two other musk-responsive subtypes that displayed a higher representation in males and females housed with males, Olfr1440 and Olfr1431 (*van der Linden et al., 2018*), showed stimulation-dependent changes in newborn OSN quantities in both adolescent male and female mice, while a third such subtype, Olfr1437, exhibited no occlusion-mediated changes in quantities of newborn OSNs in adolescent mice of either sex (*Figures 1 and 2*, and data not shown). Considering the close relationship of the ORs that define musk-responsive OSN subtypes (*McClintock et al., 2014*), these differences are intriguing. One conceivable explanation is that musk-responsive OSN subtypes vary in their sensitivity to distinct musk-like odorant molecules (*Sato-Akuhara et al., 2016*; *Horio et al., 2019*), which may be differentially emitted by mice in an age- and sex-dependent manner. Indeed, mouse odor profiles are known to vary considerably as a function of age and sex (*Osada et al., 2008*; *Osada et al., 2003*; *Schwende et al., 1986*; *Stopková et al., 2023*). In support of this, we have found that the exposure of adolescent mice to adults selectively intensifies UNO-induced open-side biases in quantities of newborn Olfr1431 OSNs. A more complete understanding of the role of odor-stimulation-dependent neurogenesis in altering the representations of distinct musk-responsive subtypes in the mouse OE will require determining the identities and sources of the natural ligands that stimulate these subtypes and how exposure to these odorants affects OSN birthrates. Future studies will also be needed to determine the mechanism that endows musk-responsive subtypes, as well as a fraction of other subtypes whose ligands have yet to be identified (*van der Linden et al., 2020*), with the capacity to undergo stimulation-dependent neurogenesis, and whether these subtypes detect odors with special functional salience.

## Materials and methods

**Key resources table**

| Reagent type (species) or resource | Designation | Source or reference | Identifiers | Additional information |
|---|---|---|---|---|
| Strain, strain background (*Mus musculus*) | C57BL/6 J | The Jackson Laboratory | Strain # 000664 | PD 28, 35, or 65 days (*Supplementary file 1*) |
| Chemical compound, drug | (*R*)–3-methylcyclopentadecanone (muscone) | Ambeed | Cat# A275816 | |
| Chemical compound, drug | 5-cyclohexadecenone (ambretone) | TCI Ltd. | Cat# C0874 | |
| Chemical compound, drug | 2-(sec-Butyl)–4,5-dihydrothiazole (SBT) | Ambeed | Cat# A578012 | |
| Chemical compound, drug | isoamyl acetate (IAA) | TCI Ltd. | Cat# A0033 | |
| Chemical compound, drug | 8-cyclohexadecenone (globanone) | Perfumer's Apprentice | Cat # ac-1510-sz1 | |

*Continued on next page*

*Continued*

| Reagent type (species) or resource | Designation | Source or reference | Identifiers | Additional information |
|---|---|---|---|---|
| Chemical compound, drug | Cyclopentadecanol was | fromTCI America | Cat# C1528 | |
| Chemical compound, drug | 5-ethynyl-2′-deoxyuridine (EdU) | Carbosynth | Cat# NE08701 | 10 mg/mL in PBS; 50 mg EdU/kg mouse |
| Commercial assay or kit | Expand High Fidelity PCR System | Roche | Cat# 11732650001 | |
| Commercial assay or kit | TOPO TA Cloning Kit, Dual Promoter | ThermoFisher | Cat# K460001 | |
| Commercial assay or kit | DIG RNA Labeling Mix | Roche | Cat# 11277073910 | |
| Peptide, recombinant protein | T7 RNA Polymerase | NEB | Cat# M0251L | |
| Peptide, recombinant protein | Sp6 RNA Polymerase | NEB | Cat# M0207S | |
| Peptide, recombinant protein | RQ1 RNase-Free DNase | Promega | Cat# M6101 | |
| Sequence-based reagent | DIG-labeled RNA fluorescent in situ hybridization probes | This paper | Please see *Supplementary file 2* | FISH (1:750) |
| Antibody | Anti-Digoxigenin-POD, Fab fragments | Roche | Cat# 11207733910, RRID:AB_514500 | FISH (1:1000) |
| Commercial assay or kit | Tyramide Signal Amplification Plus Fluorescein Kit | Akoya Biosciences | SKU NEL741001KT | |
| Chemical compound, drug | Sulfo-Cyanine3 azide | Lumiprobe | Cat# A1330 | EdU staining (4 μM) |
| Software, algorithm | Zen Blue software | Zeiss | | |
| Software, algorithm | Imaris software | Oxford Instruments | version 10.2 | |
| Other | Hibernate AB Complete medium | BrainBits | Cat# HAB100 | |
| Peptide, recombinant protein | Papain | BrainBits | Cat# PAP | |
| Other | Hibernate A-minus Calcium medium | BrainBits | Cat# HACA100 | |
| Other | NbActiv1 neuronal culturing medium | BrainBits | Cat# NbActiv1-100 | |
| Commercial assay or kit | LIVE/DEAD Fixable Aqua Dead Cell Stain Kit | Invitrogen | Cat# L34957 | |
| Commercial assay or kit | Chromium Single Cell 3' HT Kit v3.1 | 10 X Genomics | PN-1000348 | |
| Software, algorithm | Cell Ranger software | 10 X Genomics | Version 8.0.1 | |
| Software, algorithm | Loupe Browser software | 10 X Genomics | Version 8.0.0 | |
| Software, algorithm | Prism software | Graphpad | Version 10 | |

## Experimental model and subject details

All procedures involving mice were carried out in accordance with NIH standards and approved by the University of Colorado Anschutz Medical Campus Institutional Animal Care and Use Committee (IACUC). For all experiments described, tissue samples were obtained from male or female C57BL/6 J mice that were 28, 35, or 65 days of age at the time of sacrifice (*Supplementary file 1*). Except for adult-exposed mice, which remained with their parents until sacrifice, all mice were weaned at PD 21 and group-housed either sex-separated (4 females or 4 males per cage) or sex-combined (2 females and 2 males per cage) in standard cages.

## Method details
### Unilateral naris occlusion (UNO)
P14 pups were administered buprenorphine (extended release; 1 mg/kg) subcutaneously using a 31-gauge needle, anesthetized using isoflurane (completeness of anesthesia confirmed through a

tail pinch), and then immediately subjected to electrocautery for ~5 s on the right nostril under a dissecting microscope. During electrocautery, care was taken to avoid contact of the electrocautery unit with any non-superficial tissues. Pups were examined daily following the procedure to ensure complete blockage of the right nostril through scar formation (typically ~3–5 days after the procedure) and normal development and activity.

## Preparation of single cells from the open and closed sides of OEs from UNO-treated mice for single-cell RNA sequencing (scRNA-seq) analyses

Generation of the OE 2 dataset (this study) was performed as described previously for dataset OE 1 (*van der Linden et al., 2020*), with modifications. Briefly, the OE was dissected from a PD 28 male mouse that had been UNO-treated at PD 14 and weaned sex-separated at PD 21 (*Figure 1A and B*). The OE was separated at the midline using a clean scalpel, and each half was separately minced in 750 µL Hibernate AB Complete media (HAB; BrainBits, HAB100) using a razor blade. For each half, the minced tissue and HAB, along with an additional 750 µL of HAB, were transferred to a 15 mL conical tube using a 1000 µL wide-bore pipette tip and allowed to settle for 1 min, after which the HAB supernatant was removed and transferred to a new 15 mL conical tube. Minced OE halves were each suspended in 1.5 mL of papain solution (2 mg/mL papain [BrainBits; PAP] in Hibernate A-minus Calcium medium [HA-Ca; BrainBits, HACA100]) and incubated for 20 min at 37 °C, after which the papain solution was removed and discarded. To each digested OE half, the previously saved HAB supernatant was added, 750 µL at a time, and used to triturate the tissue using a 1000 µL wide-bore pipette tip with 10–15 passes to achieve ~85% tissue dissociation, after which any large pieces of tissue debris were allowed to settle for ~20 s. Suspended cells were transferred and combined into a new 15 mL conical tube, centrifuged (300 RCF; 3 min, 4 °C), resuspended in 1 mL pre-warmed neuronal culturing medium (BrainBits; NbActiv1-100), filtered using a 40 µm Flowmi cell strainer (Bel-Art; H13680-0040), quantified using trypan blue staining and hemocytometry, centrifuged (300 RCF; 3 min, 4 °C), resuspended in 1 x PBS, stained using the LIVE/DEAD Fixable Aqua Dead Cell Stain Kit (Invitrogen, L34957) according to the manufacturer's instructions, centrifuged (300 RCF; 3 min, 4 °C), resuspended in 1 mL of 1% bovine serum albumin in 1 x PBS, centrifuged (300 RCF; 3 min, 4 °C), and resuspended in 400 µL of sorting buffer (1 x PBS, 1 mM EDTA, 25 mM Hepes, pH 7, 1% FBS). Live (non-fluorescent) single cells were sorted using an Astrios Cell Sorter (Beckman Coulter) into a 1.5 mL collection tube containing 20 µL of collection buffer (10% FBS in NbActiv1 media), and used for single cell capture and sequencing.

### Single-cell capture, library preparation, and sequencing

Immediately following sorting, single cells were processed and sequenced within the Genomics Core at the University of Colorado, Anschutz Medical Campus. Following quantification *via* hemocytometry, ~19,000 cells per sample were targeted for capture using a Chromium X controller (10 X Genomics) and used to prepare scRNA-seq libraries using the 3' HT Kit (10 X Genomics) according to the manufacturer's instructions. Libraries were sequenced to a targeted depth of 40,000 reads/cell using a NovaSeq 6000 instrument (Illumina). Sequencing datasets were processed and integrated using Cell Ranger Software (version 8.0.1).

### Odorant exposure

A 1 mL aliquot of 0.1, 1, or 10% odorant solution [(R)–3-methylcyclopentadecanone (muscone; Ambeed, Inc, A275816); 5-Cyclohexadecenone (ambretone; TCI Ltd., C0874; 0.1%); 2-(sec-Butyl)–4,5-dihydrothiazole (SBT; Ambeed, Inc, A578012); or isoamyl acetate (IAA; TCI Ltd., A0033)] in propylene glycol was transferred to a compactly folded piece of absorbent paper (KimTech), which was enclosed within a metal tea ball that was hung within a standard mouse cage. Mice (4 per cage) were housed within odorant-containing cages from weaning (PD 21) until sacrifice (PD 35 or PD 65), with the odorants refreshed every other day.

### 2-Deoxy-5-ethynyluridine (EdU) injections

EdU (Carbosynth; NE08701) was administered intraperitoneally to C57BL/6 J mice at PD 28 or PD 56–58 (2 injections/day, spaced 3 hr apart, of 10 mg/mL EdU in PBS; 50 mg EdU/kg mouse body weight/injection) (*Hossain et al., 2023*).

## RNA-fluorescent in situ hybridization (RNA-FISH) probe design and generation

FISH probes were designed to span 500–1000 base pairs and were targeted to CDS and/or UTR regions of each mRNA transcript (*Supplementary file 2*). Probes were designed to minimize predicted cross-hybridization with off-target transcripts, as assessed using BLAST (NCBI) alignment to the mouse genome. To further minimize off-target cross-hybridization, probes aligning to three distinct transcript regions were generated for each target mRNA and the probe yielding the best signal strength and homogeneity was selected for use. Target probe sequences were amplified by PCR from whole OE cDNA using target-specific primers (*Supplementary file 2*), inserted into the pCRII-TOPO vector (ThermoFisher), and confirmed by Sanger sequencing. DIG-labeled antisense riboprobes were generated from 1 µg of linearized plasmid templates using T7 or Sp6 RNA polymerases (NEB) and DIG-11-UTP (Roche), treated with DNaseI (Promega) to remove the template DNA, purified via ethanol precipitation, and dissolved in 30 µL of water. A detailed description of this procedure was published previously (*Hossain et al., 2023*).

## One-color RNA fluorescent in situ hybridization (RNA-FISH) combined with EdU staining via click chemistry

OEs were dissected from experimental mice (age PD 35 or PD 65), placed in a cryomold containing OCT, flash-frozen in liquid-nitrogen-cooled isopentane, and stored at –80 °C until sectioning. Tissues were cut into 12-µm-thick cryo-sections, placed onto slides, and stored at –80 °C until staining. Slide-mounted sections were warmed (37 °C, 5 min), equilibrated in phosphate-buffered saline (PBS; pH 7.2; 3 min, room temperature [RT]), fixed in paraformaldehyde (PFA; 4% in PBS; 10 min, RT), washed in PBS (3 min, RT), permeabilized with Triton-X-100 (0.5% in PBS; 10 min, RT) followed by sodium dodecyl sulfate (1% in PBS; 3 min, RT), washed in PBS (3×3 min, RT), incubated in acetylation solution (triethanolamine [0.1 M; pH 7.5], acetic anhydride [0.25%]; 10 min, RT), washed in PBS (3×3 min, RT), incubated in hybridization solution (formamide [50%], SSC [5×], Denhardts [5×], yeast tRNA [250 µg/ mL], herring sperm DNA [500 µg/mL], heparin [50 µg/mL], EDTA [2.5 mM], Tween-20 [0.1%], CHAPS [0.25 %]; 30 min, RT), hybridized with a DIG-labeled antisense RNA probe (1:750 in hybridization solution; 16 hr, 65 °C), washed with SSC (5×; 1×5 min, 65 °C), washed with SSC (0.2×; 4×20 min, 65 °C), incubated in $H_2O_2$ (3% in TN [Tris-HCl (0.1 M; pH 7.5), 0.15 M NaCl]; 30 min, RT), washed in TNT (Tween-20 [0.05%] in TN; 5×3 min, RT), incubated in TNB (Blocking Reagent [Akoya Biosciences; 0.05% in TN]; 30 min, RT), incubated with anti-DIG-POD antibody (Roche; 1:1000 in TNB; 12 hr, 4 °C), and washed in TNT (3×20 min, RT). Fluorescent signals corresponding to the target transcript were generated using the Tyramide Signal Amplification (TSA) Plus Fluorescein Kit (Akoya Biosciences) according to the manufacturer's instructions. Slides were washed in 3% BSA in PBS (2×5 min, RT, with gentle rocking), incubated with EdU reaction solution (4 mM $CuSO_4$, 4 µM Sulfo-Cyanine 3 Azide [Lumiprobe], 100 mM sodium ascorbate [prepared fresh], in PBS; 30 min, RT, in darkness), and washed with 3% BSA in PBS (2×3 min, RT). Slides were washed in TNT (2×3 min, RT), incubated in DAPI (300 nM in TN; 3 min, RT), washed in TNT (1×3 min, RT), and mounted using Vectashield (Vector Laboratories). A detailed description of this procedure was published previously (*Hossain et al., 2023*).

### Image acquisition and processing

Images were acquired using a Zeiss LSM 900 with Airyscan 2 microscope with an automated stage and Zen Blue software (Zeiss). Mosaic images were stitched, and each fluorescence channel was adjusted individually to enhance contrast using Zen Blue software. Representative images were exported in jpg format and rotated and cropped using Adobe Photoshop. A detailed description of this procedure was published previously (*Hossain et al., 2023*).

### Quality criteria for sectioned OEs

For UNO-treated mice, UNO efficiency was assessed by visual inspection of mouse nostrils prior to dissection. Additionally, OE cryosections from UNO-treated mice were stained *via* one-color RNA-FISH for *S100a5* transcript levels. For each section analyzed, *S100a5* mRNA intensities were evaluated within paired regions on the two sides of each OE section. OEs from UNO-treated mice were excluded from further analysis if the mean fluorescence intensity corresponding to *S100a5* transcripts

was not noticeably (~10-fold) greater on the open side of the OE compared to the closed side. All sections from both UNO-treated and non-occluded mice were assessed for left-right symmetry and intactness. Individual OE sections were excluded if they lacked a high degree of symmetry or were less than 90% intact. No sections were otherwise excluded. A detailed description of this procedure was published previously (*Hossain et al., 2023*).

## Gas chromatography – mass spectrometry (GC-MS) analysis of mouse preputial gland (PG) extracts

Immediately following euthanasia, three PGs were dissected from 5-week-old male mice, transferred to separate 1.5 mL microcentrifuge tubes, and flash-frozen on dry ice. Frozen PG tissues were shipped on dry ice to the Analytical Resource Core at Colorado State University for processing, essentially as described previously (*Bala and Babbar, 2023*). Each gland was ground in liquid nitrogen using a mortar and pestle, transferred to a glass extraction vial containing 200 µL of dichloromethane, vortexed and sonicated for 30 min, and centrifuged at $3000 \times g$ for 1 min. Supernatants were transferred to glass autosampler vials and stored at –20 °C until analysis. For GC-MS analyses, 1 µL of dichloromethane extract was injected into a Trace 1310 gas chromatography instrument coupled to a Thermo ISQ-LT mass spectrometer, at a 5:1 split ratio. The inlet was held at 285 °C. Separation was achieved on a 30 m DB-5MS column (J&W, 0.25 mm ID, 0.25 µm film thickness). Oven temperature was held at 80 °C for 0.5 min, ramped at 15 °C/min to 330 °C, and held at 330 °C for 8 min. Helium carrier gas flow was held at 1.2 mL/min. Temperatures of the transfer line and ion source were held at 300 °C and 260 °C, respectively. Full scan mode (50–650 m/z) was used on all the samples. The injection order of samples was randomized. Standards: (R)–3-methylcyclopentadecanone (muscone) was purchased from Ambeed, Inc (A275816), 5-cyclohexadecenone (ambretone) was from TCI America (C0874), 8-cyclohexadecenone (globanone) was from Perfumer's Apprentice (ac-1510-sz1), and cyclopentadecanol was from TCI America (C1528). GC-MS data were processed using Chromeleon 7.2.10 software (Thermo Fisher Scientific).

## Quantification and statistical analysis

### scRNA-seq analysis of UNO-induced changes in subtype-specific quantities of newborn OSNs

Newborn OSNs of specific subtypes were quantified within scRNA-seq datasets generated from the open and closed sides of OEs from two different male C57BL/6 J mice that were UNO-treated at PD 14 and sacrificed at PD 28 (*Figure 1*): OE 1 (generated previously *van der Linden et al., 2020*; https://www.ncbi.nlm.nih.gov/geo/query/acc.cgi?acc=GSE157119), and OE 2 (this study; https://www.ncbi.nlm.nih.gov/geo/query/acc.cgi?acc=GSE278693). Newborn OSNs of specific subtypes were identified by their expression of the iOSN-specific marker *Gap43* (Log$_2$ UMI >1) and a specific OR gene (Log$_2$ UMI >2; *Figure 1*) and quantified within the open and closed sides of the OE 1 and OE 2 datasets using Loupe Browser software (8.0.0; 10 X Genomics). Subtype-specific quantities of newborn OSNs in each OE half were normalized based on the total number of cells or OSNs (*Gap43* Log$_2$ UMI >1 or *Omp* Log$_2$ UMI >3) within the corresponding dataset.

### scRNA-seq analysis of OR transcripts within individual INP3, iOSN, and mOSN cells on the open and closed sides of OEs from UNO-treated mice

Individual INP3, iOSN, and mOSN cells expressing transcripts of a specific OR gene were identified based on their expression of *Tex15* (*Fletcher et al., 2017*; *Pourmorady et al., 2024*) (Log$_2$ UMI >2) and the OR (Log$_2$ UMI >0), their expression of *Gap43* (Log$_2$ UMI >1) and the OR (Log$_2$ UMI >2), or their expression the OR (Log$_2$ UMI >2) and lack of expression of *Gap43* (Log$_2$ UMI ≤1), respectively, using Loupe Browser software (8.0.0). The cellular transcript levels of all ORs within INP3, iOSN, and mOSN cells expressing a specific musk-responsive (Olfr235, Olfr1440) or non-musk responsive (Olfr701) OR within the open and closed datasets of OE 1 and OE 2 samples were calculated and exported using Loupe Browser software (8.0.0).

## Histological quantification of newborn and total OSNs of specific subtypes

Quantities of newborn and total OSNs within the OEs of individual mice were determined from images of a series of at least 5 coronal tissue sections stained for EdU and a specific OR mRNA and spanning the anterior-posterior length of each OE. Quantifications were performed separately on the two sides of each OE section, with the experimenter blinded to the sample identities. Newborn cells of specific subtypes were identified based on robust nuclear EdU staining (Cy3; >2 fold above background) that was overlapped at least 50% by OR mRNA staining (FITC). A detailed description of this procedure was published previously (*Hossain et al., 2023*).

## Comparison of methods for normalizing quantities of newborn OSNs of specific subtypes

For all histological quantifications of newborn and total OSNs of specific subtypes in this study, OSN quantities were normalized to the number of half-sections analyzed. This approach has been used in multiple previous studies for quantifying newborn (OR+/EdU+) and total (OR+) OSN abundances (*Ibarra-Soria et al., 2017*; *van der Linden et al., 2018*; *van der Linden et al., 2020*; *Hossain et al., 2023*). To verify the rigor of this method, newborn Olfr235 OSN quantities were also normalized by total quantities of newborn OSNs (approximated by total EdU+ cells) or total quantities of cells (approximated by DAPI+ area) for three experimental comparisons in this study: the open versus the closed sides of OEs from UNO-treated male mice (*Figure 2—figure supplement 2*), OEs from non-occluded females versus males (*Figure 3—figure supplement 2*), and OEs from non-occluded females exposed just to female littermates versus females also exposed to 0.1% muscone (*Figure 4—figure supplement 4*). For each condition, EdU+ cells and DAPI+ areas were identified and quantified within images of OE sections stained for Olfr235 mRNA and EdU using the surface and stats tools, respectively, within Imaris software (version 10.2).

## Statistics

To assess deviation from normality of the histological quantifications of newborn and total OSNs of specific subtypes used for comparisons in this study, all datasets were tested using the Shapiro-Wilk test for non-normality and the p values obtained are included within Figure Source Data files. Of the 274 datasets tested, 253 have Shapiro-Wilk p-values >0.05, indicating that the vast majority (92%) do not show evidence of significant deviation from a normal distribution. A general lack of deviation of the datasets in this study from a normal distribution is further supported by quantile-quantile (QQ) plots (*Appendix 1—figure 3*). Moreover, results of both parametric and non-parametric statistical tests of comparisons in this study are, in general, in good agreement (*Supplementary file 3*). Statistical analyses of differences in OSN quantities between the open and closed sides of OEs from UNO-treated mice were performed using ratio paired two-tailed *t*-tests (parametric) and Wilcoxon matched-pairs signed rank two-tailed tests (non-parametric). For these tests, the open and closed sides of each OE were paired, enabling testing of differences between the two sides independent of OSN number and staining variance between sections. Statistical analyses of differences in OSN quantities in the OEs of non-occluded mice or UNO effect sizes in UNO-treated mice subjected to two different experimental conditions were performed using unpaired two-tailed *t*-tests (parametric) and Mann-Whitney two-tailed tests (non-parametric). Statistical analyses of differences in OSN quantities in the OEs of non-occluded mice or UNO effect sizes in UNO-treated mice subjected to more than two different experimental conditions were performed using one-way ANOVA tests, FDR-adjusted using the two-stage linear step-up procedure of Benjamini, Krieger and Yekutieli (parametric) and Kruskal-Wallis tests, FDR-adjusted using the two-stage linear step-up procedure of Benjamini, Krieger and Yekutieli (non-parametric). For comparisons of differences in quantities of newborn OSNs of musk-responsive subtypes at 4 and 7 days post-EdU between non-occluded mice exposed and unexposed to muscone, a two-sample ANOVA - fixed-test, using F distribution (right-tailed) was used. Data presented in figures represent mean +/-SEM. For all statistical analyses, a significance threshold of $p<0.05$ was used. All statistical tests were performed using Prism 10 software (Graphpad). Results of all statistical tests performed in this study are summarized in *Supplementary file 3*. p values reported within the text and figures are based on parametric tests.

## Sample-size estimation

Results from previous studies (*van der Linden et al., 2018*; *van der Linden et al., 2020*) were used to determine an appropriate sample size for comparing the number of total (OR+) and newborn (OR+/EdU+) OSNs on the open and closed sides of the OE. Previously, it was found that for an OR with a typical expression frequency (~0.1%) and an effect size of ~twofold, 12 OE sections taken from four different animals were sufficient to find a highly statistically significant difference (p<0.001; paired two-tailed *t* test). For comparisons between different animals, results from previous studies (*van der Linden et al., 2018*; *van der Linden et al., 2020*) were also used to determine an appropriate sample size. Previously, we had found that for an OR with a typical expression frequency (~0.1%) and an effect size of ~twofold, 20 OE sections taken from four different animals was sufficient to find a highly statistically significant difference between different animals (p<0.01; two-tailed unpaired *t* test).

## Acknowledgements

We are grateful to members of the Santoro lab for helpful discussions and for comments and suggestions on this manuscript. This article is based upon work supported by the National Science Foundation (Grant No. 1943528) and the National Institutes of Health (NIDCD; R01DC019936).

# Additional information

### Funding

| Funder | Grant reference number | Author |
| --- | --- | --- |
| National Institute on Deafness and Other Communication Disorders | R01DC019936 | Stephen W Santoro |
| National Science Foundation | 1943528 | Stephen W Santoro |

The funders had no role in study design, data collection and interpretation, or the decision to submit the work for publication.

### Author contributions

Kawsar Hossain, Conceptualization, Data curation, Formal analysis, Investigation, Methodology, Writing - original draft; Madeline Smith, Formal analysis, Investigation; Karlin E Rufenacht, Investigation; Rebecca O'Rourke, Data curation; Stephen W Santoro, Conceptualization, Data curation, Formal analysis, Supervision, Funding acquisition, Investigation, Methodology, Writing - original draft, Project administration, Writing - review and editing

### Author ORCIDs

Rebecca O'Rourke ⓘ https://orcid.org/0000-0003-1198-6963
Stephen W Santoro ⓘ https://orcid.org/0000-0003-1870-2513

### Ethics

All procedures involving mice were carried out in accordance with NIH standards and approved by the University of Colorado Anschutz Medical Campus Institutional Animal Care and Use Committee (protocol #01023).

Reviewer #1 (Public review): https://doi.org/10.7554/eLife.96152.3.sa1
Reviewer #3 (Public review): https://doi.org/10.7554/eLife.96152.3.sa2
Author response https://doi.org/10.7554/eLife.96152.3.sa3

## Additional files

### Supplementary files

Supplementary file 1. Summary of experimental conditions and numbers of biological replicates (mice) tested for each condition.

Supplementary file 2. RNA-FISH probes used in this study.

Supplementary file 3. Summary of statistical analyses.

MDAR checklist

### Data availability

Sequencing data generated for this study have been deposited in GEO under accession code GSE278693. All data generated during this study are included in the manuscript and supporting files; source data files have been provided for all figures and figure supplements.

The following dataset was generated:

| Author(s) | Year | Dataset title | Dataset URL | Database and Identifier |
|---|---|---|---|---|
| Santoro SW, Rufenacht KE, O'Rourke R, Hossain K | 2025 | scRNA-seq analysis of the open and closed sides of the mouse olfactory epithelium following unilateral naris occlusion | https://www.ncbi.nlm.nih.gov/geo/query/acc.cgi?acc=GSE278693 | NCBI Gene Expression Omnibus, GSE278693 |

The following previously published dataset was used:

| Author(s) | Year | Dataset title | Dataset URL | Database and Identifier |
|---|---|---|---|---|
| Santoro SW | 2020 | scRNA-seq of the open and closed sides of the mouse olfactory epithelium following unilateral naris occlusion | https://www.ncbi.nlm.nih.gov/geo/query/acc.cgi?acc=GSE157119 | NCBI Gene Expression Omnibus, GSE157119 |

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

## Appendix 1

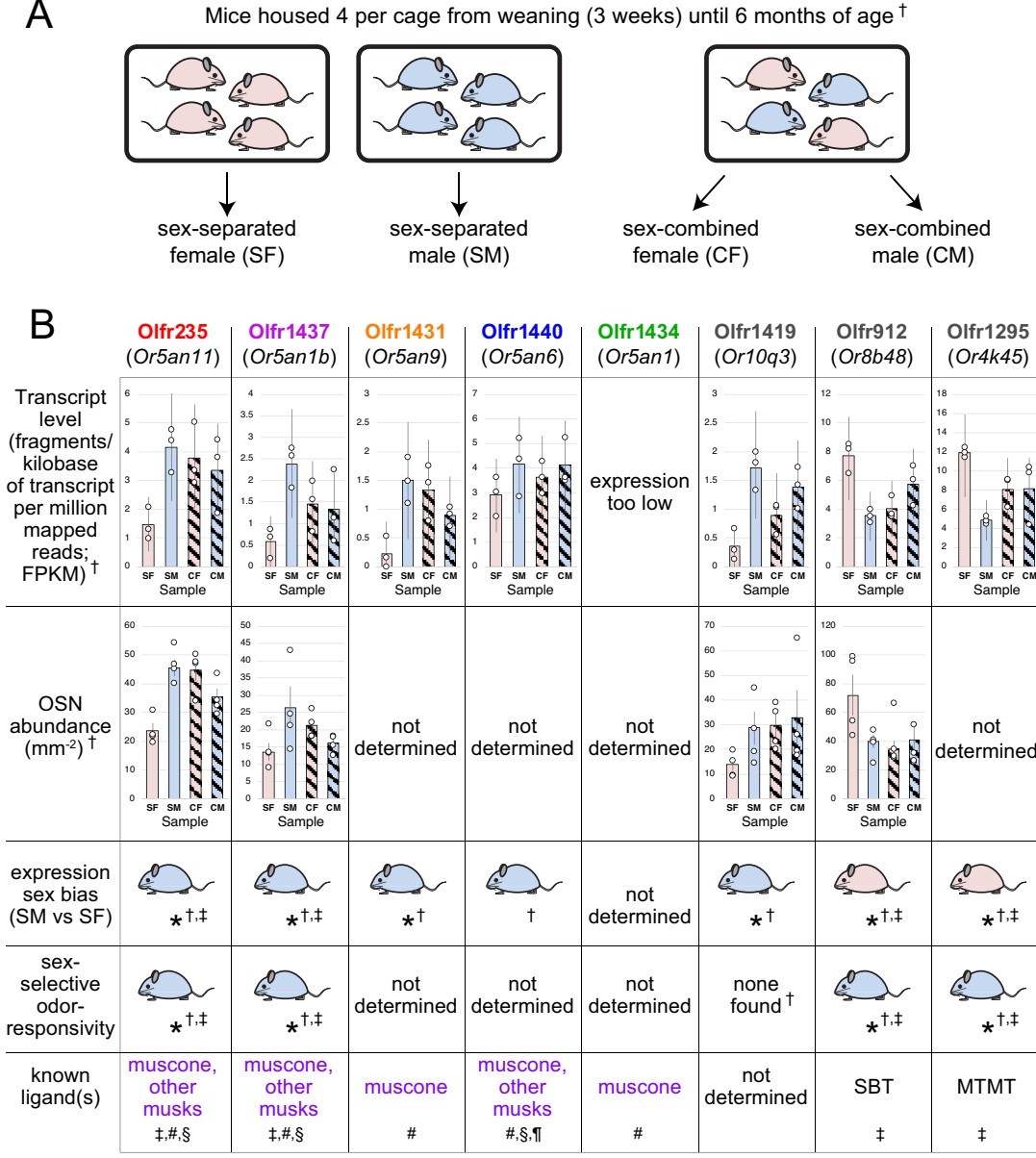

**Appendix 1—figure 1.** Identification of OSN subtypes that are candidates for undergoing sex-specific- and/or musk odor-accelerated neurogenesis. (**A**) In a previous study, the OE transcript profiles of female and male mice that were housed either sex-separated or sex-combined from weaning (PD 21) until 6 months of age were profiled and compared *via* bulk RNA-seq (***van der Linden et al., 2018***; ***Santoro and Jakob, 2018***). (**B**) OSN subtypes previously identified as responsive to sex-specific odors and/or musk-like odors. SBT, 2-sec-butyl-4,5-dihydrothiazole; MTMT, (methylthio)methanethiol; †, ***van der Linden et al., 2018***; ***Santoro and Jakob, 2018***; ‡, ***Vihani et al., 2020***; #, ***McClintock et al., 2014***; §, ***Sato-Akuhara et al., 2016***; ¶, ***Shirasu et al., 2014***. Error bars: 95% confidence intervals.

A

| Predicted matches to known musks | Retention time (RT) | Predicted potential match | RT difference from standard |
|---|---|---|---|
| Unknown 1 | 11.55 | 8-cyclohexadecenone | 0.06, 0.11 |
| Unknown 2 | 12.27 | cycloheptadecanol | not determined |
| Unknown 3 | 15.38 | cyclopentadecanol | 4.48 |

B

| Standards tested | Retention time |
|---|---|
| (R)-3-methylcyclopentadecanone (muscone) | 10.98 |
| 5-cyclohexadecenone (ambretone) | 11.45 |
| 8-cyclohexadecenone (globanone) | 11.44, 11.49 |
| cyclopentadecanol (normuscol) | 10.90 |

C

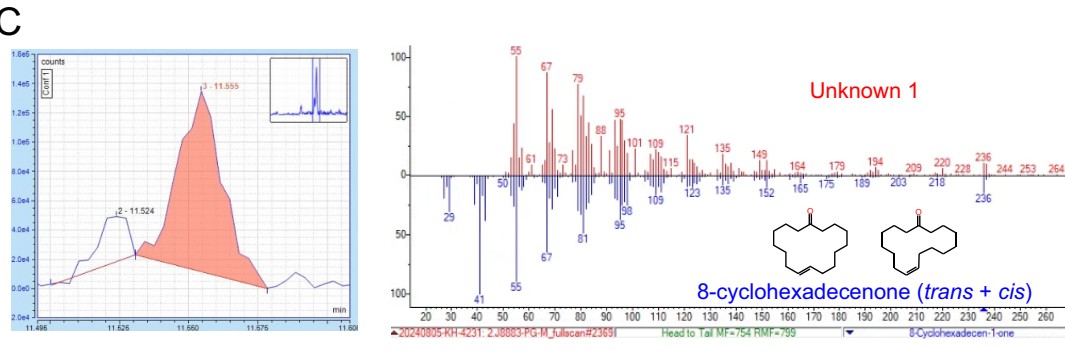

D

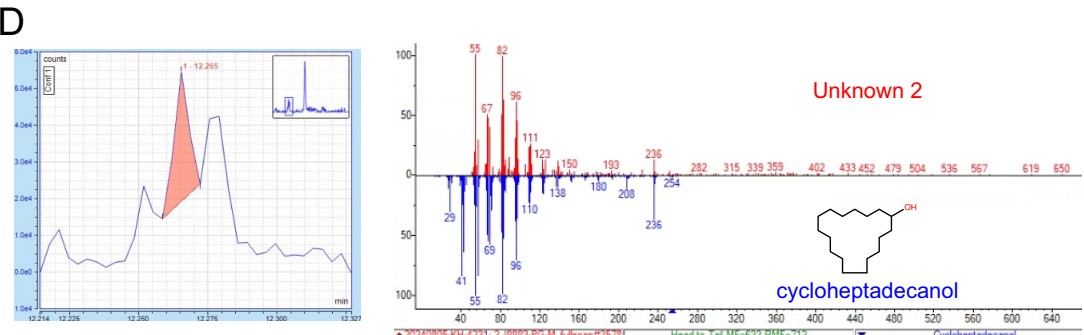

**Appendix 1—figure 2.** Gas chromatography – mass spectrometry (GC-MS) analyses of mouse preputial gland extracts for molecules with structural similarity to known musk odorants. (**A**, **B**) GC-MS signals from male mouse preputial gland extracts (**A**) and commercially available samples of 4 known musk compounds (**B**). Analyses of preputial glands revealed signals (Unknowns 1–3) with predicted potential matches to the indicated odorants based on spectral similarities (**A**). Experimental differences in retention times between unknowns and standards are indicated. (**C**) *Left*: Region of the extracted ion chromatograph (m/z 236) of a preputial gland extract, with the signal corresponding to Unknown 1 highlighted in red. *Right*: Mass spectra corresponding to Unknown 1 (*red*) and a predicted match, 8-cyclohexadecenone (*blue*), a musk compound that was previously found to activate Olfr235 and Olfr1440 (**Sato-Akuhara et al., 2016**). The retention times of Unknown 1 and 8-cyclohexadecenone differ by 0.06 and 0.11 minutes (possibly corresponding to the *cis* and *trans* isomers) (**A**), indicating potential structural similarity. (**D**) *Left*: Region of the extracted ion chromatograph (m/z 236) of a preputial gland extract, with the signal corresponding to Unknown 2 highlighted in red. *Right*: Mass spectra corresponding to Unknown 2 (*red*) and a predicted match, cycloheptadecanol (*blue*). Confirmation of a match will require comparison of the observed retention time of Unknown 2 (12.27 minutes) with that of a cycloheptadecanol standard (not determined). Although Unknown 3 exhibited a predicted match to cyclopentadecanol, a musk compound previously found to activate Olfr235 and Olfr1440 (**Sato-Akuhara et al., 2016**), the observed retention time difference of 4.48 minutes (**A**) indicates substantial structural dissimilarity.

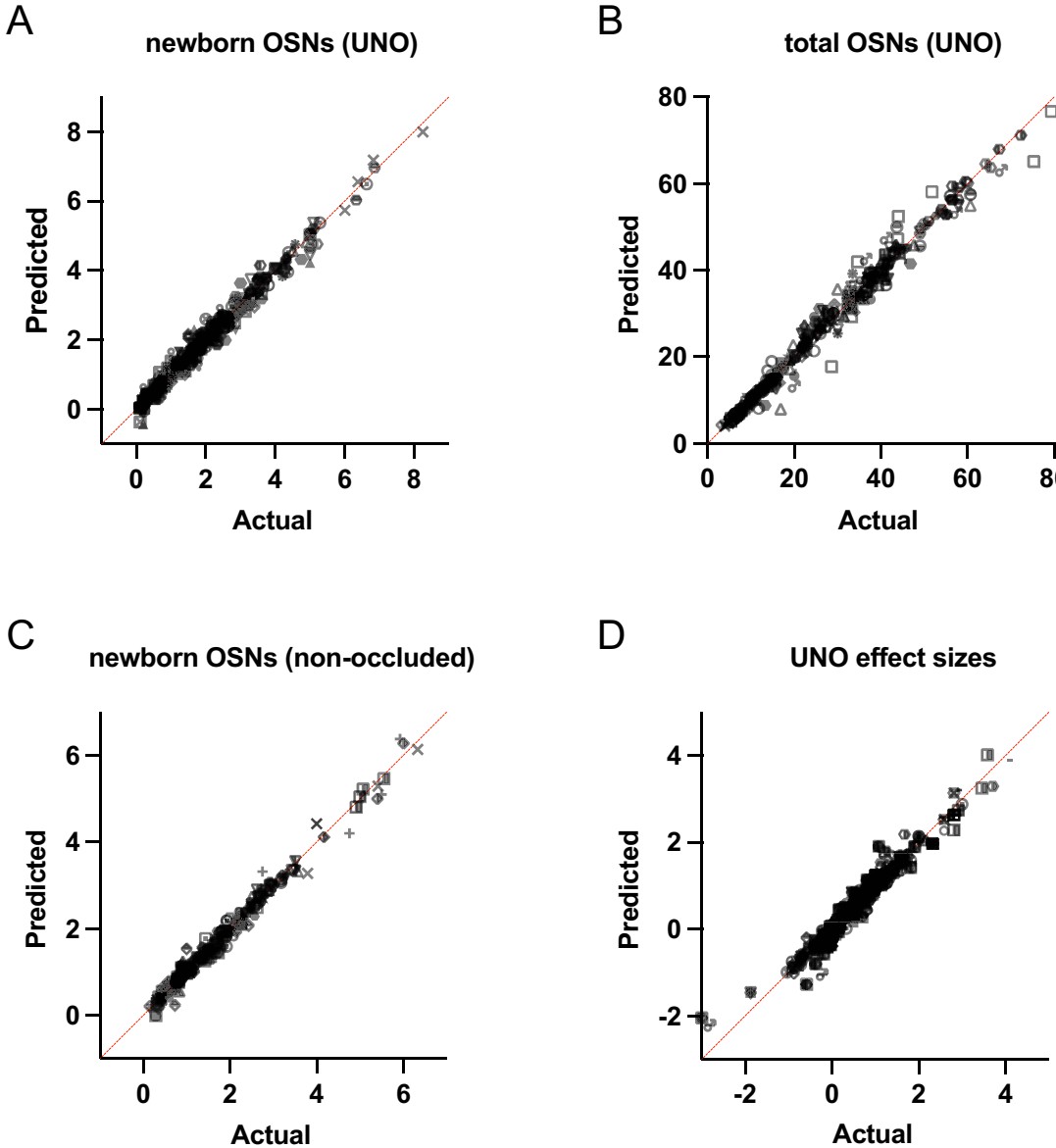

**Appendix 1—figure 3.** Quantile-quantile (QQ) plots for comparison of actual data to a theoretically normal distribution. (**A**) Analysis of quantities of newborn OSNs on the open and closed sides of OEs from UNO-treated mice. (**B**) Analysis of quantities of total OSNs on the open and closed sides of OEs from UNO-treated mice. (**C**) Analysis of quantities of newborn OSNs within OEs from non-occluded mice. (**D**) Analysis of UNO effect sizes of newborn and total OSNs on the open and closed sides of OEs from UNO-treated mice. In each plot, predicted values (assuming sampling from a Gaussian distribution) are plotted on the vertical axis, while actual values are plotted on the horizontal axis. Points generally follow the line of identity, indicating that the data reflect a Gaussian (normal) distribution.

