## [Editor Report · eLife Assessment]

This study provides **valuable** insights into the role of sensory stimulation in neurogenesis in the mammalian olfactory epithelium, where new olfactory sensory neurons are continually born throughout an animal's lifespan. The authors show that exposure to two different musk-related odors specifically increases the birth rates of those neurons that respond to these odors. This potentially results in adaptive changes in the subtype composition of the olfactory sensory neuron population. **Solid** evidence, well supported by control experiments, is presented to support these findings, though further work is needed to confirm that this phenomenon generalizes to olfactory sensory neurons expressing other types of odorant receptor and to explore the mechanisms underlying the stimulus specificity of neurogenesis.

---

## [Referee Report · Reviewer #1 (Public review)]

Summary

Olfactory sensory neurons (OSNs) in the olfactory epithelium detect myriads of environmental odors that signal essential cues for survival. OSNs are born throughout life and thus represent one of the few neurons that undergo life-long neurogenesis. Until recently, it was assumed that OSN neurogenesis is strictly stochastic with respect to subtype (i.e. the receptor the OSN chooses to express). However, a recent study showed that olfactory deprivation via naris occlusion selectively reduced birthrates of only a fraction of OSN subtypes and indicated that these subtypes appear to have a special capacity to undergo changes in birthrates in accordance with the level of olfactory stimulation. These previous findings raised the interesting question of what type of stimulation influences neurogenesis, since naris occlusion does not only reduce the exposure to potentially thousands of odors, but also to more generalized mechanical stimuli via preventing airflow.

In this study, the authors set out to identify the stimuli that are required to promote the neurogenesis of specific OSN subtypes. Specifically, they aim to test the hypothesis if discrete odorants selectively stimulate the same OSN subtypes whose birthrates are affected. This would imply a highly specific mechanism in which exposure to certain odors can "amplify" OSN subtypes responsive to those odors suggesting that OE neurogenesis serves, in part, an adaptive function.

To address this question, the authors focused on a family of OSN subtypes that had previously been identified to respond to musk-related odors and that exhibit higher transcript levels in the olfactory epithelium of mice exposed to males compared to mice isolated from males. First, the authors confirm via a previously established cell birth dating assay in unilateral naris occluded mice that this increase in transcript levels actually reflects a stimulus-dependent birthrate acceleration of this OSN subtype family. In a series of experiments (in unilateral occluded and non-occluded mice) using the same birth dating assay, they show that several subtypes of this OSN family, but not other "control" subtypes exhibit increased birthrates in response to adolescent male exposure, but not to female exposure.

In the core experiment of the study, they expose unilaterally naris occluded and non-occluded mice to two musk-related odors and two "control" odors (that do not activate musk-responsive OSN subtypes) to test if these odors specifically accelerate the birth rates of OSN types that are responsive to these odors. This experiment reveals that (for the tested odors and OSN subtypes) indeed birthrates are only affected by discrete odorants that stimulate these OSN subtypes (with a complex relationship between birth rate acceleration and odor concentrations) suggesting that OE neurogenesis may serve, in part, as an adaptive function

Strength:

The scientific question is valid and opens an interesting direction. The previously established cell birth dating assay in naris occluded and non-occluded mice is well performed and accompanied by several control experiments addressing potential other interpretations of the data.

In this revised version, the authors added several new experiments addressing the previous concern that only the effect of one specific odor (muscone) on musk-responsive OSN subtypes had been tested to make the general claim that discrete odors specifically accelerate the birth rate of OSN subtypes they stimulate. Now the authors demonstrate that another musk-related odor (ambretone) also induces this effect and that other non-musk odors do not. In addition, they show that two other OSN subtypes that do not respond to musk-related odors are not affected. These experiments further substantiate the above claim.

Weakness:

(1) The main research question of this study was to test if discrete odors specifically accelerate the birth rate of OSN subtypes they stimulate, i.e. does muscone only accelerate the birth rate of OSNs that express muscone-responsive ORs, or vice versa is the birthrate of muscone-responsive OSNs only accelerated by odors they respond to?

As mentioned under "strength" the authors added several experiments to further substantiate their claim. While these controls are very important to show that the observed effect is indeed specific for musk-related odors on musk-responsive OSN subtypes, these experiments still only focus on one closely related family of musk-responsive OSN subtypes. To understand if this phenomenon is a more generalized mechanism and plays a role for other OSN subtypes beyond this small family of related receptors, further experiments showing this effect for other OSN subtypes are critical.

(2) Previous concerns (#2, #4, #5 and #6) about a lack of increase in UNO effect size for olfr1440 under any muscone concentrations, strong fluctuations of newborn neurons on the closed side as well as a seemingly contradicting statement that overstimulation possibly reflects reduced survival have been addressed by adding potential explanations to the text.

In addition, the previous remark (#3) that certain phrases gave the misleading impression that musk-related odors are indeed excreted into male mouse urine at certain concentrations was addressed not only by re-phrasing, but by performing additional experiments. Although these did not deliver clear results (because of technical difficulties), interesting possibilities are discussed.

---

## [Referee Report · Reviewer #3 (Public review)]

Summary

Neurogenesis in the mammalian olfactory epithelium continues throughout the animal's lifespan, replacing damaged or dying olfactory sensory neurons. It has been tacitly assumed that the replacement of olfactory receptor (OR) subtypes occurs randomly. However, anecdotal evidence has suggested otherwise. In this study, Santoro and colleagues challenge this assumption by systematically exploring three key questions: Is there enrichment of specific OR subtypes during neurogenesis? Is this enrichment influenced by sensory stimuli? Does enrichment result from differential generation of OR types or differential cell death regulated by neural activity? The authors present convincing evidence that muscone stimulus selectively enhances the neurogenesis of OSNs expressing muscone receptors, suggesting that the selection of ORs in regenerating neurons is not random.

Strengths

This study is the first in formulating and systematically testing the selective promotion hypothesis. It is a comprehensive and systematic examination of multiple musk receptors under conditions of unilateral naris occlusion and various stimuli. The controls are properly done. The quality of in situ hybridization and immunofluorescent staining is high, allowing the authors to effectively estimates the number of OR types. The data convincingly demonstrate that increased expression of musk receptors in response to male odor or muscone stimulation.

Weaknesses

The revised version has addressed most initial weaknesses. However, some inconsistencies remain, raising questions about the proposed model. For instance, in the unilateral naris occlusion experiment, although average expression of non-musk receptors shows no significant change, receptors seem to fall into two groups with one subset increasing and another decreasing (Fig. 1E). This suggests naris occlusion may regulate OR expression independently of odor-induced activity, a possibility that remains unaddressed.

There is curiosity regarding receptors for the male odor SBT, olfr912, and olfr1295, which exhibit increased expression on the occluded side. The explanation that SBT exposure shortens these neurons' lifespan lacks substantiation and is inconsistent with later data. For example, Figure 3-figure supplement 3 I does not show olfr912 changes on the UNO side, and Figure 4-figure supplement 4 shows no significant decrease in olfr912 expression in SBT-exposed mice. Additionally, single-cell RNASeq experiments did not examine olfr912 and olfr1295 expression.

While the study convincingly demonstrates muscone's selective stimulation of olfr235-expressing OSNs, it lacks exploration of the mechanisms underlying this specificity. The discussion of signaling pathways remains too generic.

In summary, while the study offers significant insights into neurogenesis and OR subtype enrichment, further investigation into underlying mechanisms and addressing existing inconsistencies would strengthen its conclusions.

---

## [Author Response]

The following is the authors’ response to the original reviews

**Reviewer #1 (Public Review):**
Summary:Olfactory sensory neurons (OSNs) in the olfactory epithelium detect myriads of environmental odors that signal essential cues for survival. OSNs are born throughout life and thus represent one of the few neurons that undergo life-long neurogenesis. Until recently, it was assumed that OSN neurogenesis is strictly stochastic with respect to subtype (i.e. the receptor the OSN chooses to express).However, a recent study showed that olfactory deprivation via naris occlusion selectively reduced birthrates of only a fraction of OSN subtypes and indicated that these subtypes appear to have a special capacity to undergo changes in birthrates in accordance with the level of olfactory stimulation. These previous findings raised the interesting question of what type of stimulation influences neurogenesis, since naris occlusion does not only reduce the exposure to potentially thousands of odors but also to more generalized mechanical stimuli via preventing airflow.In this study, the authors set out to identify the stimuli that are required to promote the neurogenesis of specific OSN subtypes. Specifically, they aim to test the hypothesis that discrete odorants selectively stimulate the same OSN subtypes whose birthrates are affected. This would imply a highly specific mechanism in which exposure to certain odors can "amplify" OSN subtypes responsive to those odors suggesting that OE neurogenesis serves, in part, an adaptive function.To address this question, the authors focused on a family of OSN subtypes that had previously been identified to respond to musk-related odors and that exhibit higher transcript levels in the olfactory epithelium of mice exposed to males compared to mice isolated from males. First, the authors confirm via a previously established cell birth dating assay in unilateral naris occluded mice that this increase in transcript levels actually reflects a stimulus-dependent birthrate acceleration of this OSN subtype family. In a series of experiments using the same assay, they show that one specific subtype of this OSN family exhibits increased birthrates in response to juvenile male exposure while a different subtype shows increased birthrates to adult mouse exposure. In the core experiment of the study, they finally exposed naris occluded mice to a discrete odor (muscone) to test if this odor specifically accelerates the birth rates of OSN types that are responsive to this odor. This experiment reveals a complex relationship between birth rate acceleration and odor concentrations showing that some muscone concentrations affect birth rates of some members of this family and do not affect two unrelated OSN subtypes.

In addition to the results nicely summarized by the reviewer, which focus on experiments to examine the effects of odor stimulation on unilateral naris occluded (UNO) mice, an important part of the present study are experiments on non-occluded (i.e., non-UNO-treated) mice. These experiments show: (1) that the exposure of non-occluded mice to odors from adolescent male mice selectively increases quantities of newborn OSNs of the musk-responsive subtype Olfr235 (Figure 3G, H; previously Figure 6), (2) the exposure of non-occluded female mice to 2 different musk odorants (muscone, ambretone) selectively increases quantities of newborn OSNs of 3 musk responsive subtypes: Olfr235, Olfr1440 and Olfr1431 (Figure 4D-F; previously Figure 6), and (3) the exposure of non-occluded adult female mice to a musk odorants selectively increases quantities of newborn OSNs of musk responsive subtypes (Figure 5; previously Fig. S7). We have reorganized the revised manuscript to more prominently and clearly present the experimental design and findings of these experiments. We have also made changes to clarify (via schematics) the experimental conditions used (i.e., UNO, non-UNO, odor exposure) in each experiment.

Strengths:The scientific question is valid and opens an interesting direction. The previously established cell birth dating assay in naris occluded mice is well performed and accompanied by several control experiments addressing potential other interpretations of the data.Weaknesses:(1) The main research question of this study was to test if discrete odors specifically accelerate the birth rate of OSN subtypes they stimulate, i.e. does muscone only accelerate the birth rate of OSNs that express muscone-responsive ORs, or vice versa is the birthrate of muscone-responsive OSNs only accelerated by odors they respond to?This question is only addressed in Figure 5 of the manuscript and the results only partially support the above claim. The authors test one specific odor (muscone) and find that this odor (only at certain concentrations) accelerates the birth rate of some musk-responsive OSN subtypes, but not two other unrelated control OSN subtypes. This does not at all show that musk-responsive OSN subtypes are only affected by odors that stimulate them and that muscone only affects the birthrate of musk-responsive OSNs, since first, only the odor muscone was tested and second, only two other OSN subtypes were tested as controls, that, importantly, are shown to be generally stimulus-independent OSN subtypes (see Figure 2 and S2).As a minimum the authors should have (a) tested if additional odors that do not activate the three musk-responsive subtypes affect their birthrate (b) choose 2-3 additional control subtypes that are known to be stimulus-dependent (from their own 2020 study) and test if muscone affects their birthrates.

We appreciate these suggestions. Within the revised manuscript, we have described and included the results from several new experiments:

(1) As noted by the reviewer, we had previously tested the effects of exposure to only one exogenous musk odorant, muscone, on quantities of newborn OSNs of the musk-responsive subtypes Olfr235, Olfr1440, and Olfr1431. To test whether the effects observed with muscone exposure occur with other musk odorants, we assessed the effects of exposure to ambretone (5-cyclohexadecenone), a musk odorant previously found to robustly activate musk-responsive OSNs (Sato-Akuhara et al., 2016; Shirasu et al., 2014), on quantities of newborn OSNs of 3 musk-responsive subtypes Olfr235, Olfr1440, and Olfr1431, as well as the SBT-responsive subtype Olfr912, in the OEs of non-occluded female mice. Exposure to ambretone was found to significantly increase quantities of newborn OSNs of all 3 musk-responsive subtypes (Figure 4D-F) but not the SBT-responsive subtype (Figure 4–figure supplement 4C-left), indicating that a variety of musk odorants can accelerate the birthrates of musk responsive subtypes.

(2) To verify that exogenous non-musk odors do not increase quantities of newborn OSNs of musk responsive OSN subtypes (point a, above), we quantified newborn OSNs of 3 musk-responsive subtypes, Olfr235, Olfr1440, and Olfr1431, in non-occluded female mice that were exposed to the non-musk odorants SBT or IAA. As expected, neither of these odorants significantly affected the birthrates of the subtypes tested (Figure 4D-F).

(3) To confirm that exogenous musk odors do not accelerate the birthrates of non-musk responsive OSN subtypes that were previously found to undergo stimulation-dependent neurogenesis (point b, above), we quantified newborn OSNs of 2 such subtypes, Olfr827 and Olfr1325, in non-occluded female mice that were exposed to muscone. As expected, exposure to muscone did not significantly affect the birthrates of either of these subtypes (Figure 4–figure supplement 4C-middle, right).

(4) To provide additional confirmation that only some OSN subtypes have a capacity to exhibit increases in newborn OSN quantities in the presence of odors that activate them, we compared quantities of newborn OSNs of the SBT-responsive subtype Olfr912 in non-occluded females that were either exposed to 0.1% SBT versus unexposed controls. As expected, exposure of SBT caused no significant increase in quantities of newborn Olfr912 OSNs (Figure 4–figure supplement 4C-left).

(2) The finding that Olfr1440 expressing OSNs do not show any increase in UNO effect size under any muscone concentration (Figure 5D, no significance in line graph for UNO effect sizes, middle) seems to contradict the main claim of this study that certain odors specifically increase birthrates of OSN subtypes they stimulate. It was shown in several studies that olfr1440 is seemingly the most sensitive OR for muscone, yet, in this study, muscone does not further increase birthrates of OSNs expressing olfr1440. The effect size on birthrate under muscone exposure is the same as without muscone exposure (0%).

In contrast, the supposedly second most sensitive muscone-responsive OR olfr235 shows a significant increase in UNO effect size between no muscone exposure (0%) and 0.1% as well as 1% muscone.

Findings that quantities of newborn Olfr1440 OSNs do not show a significantly greater UNO effect size in the OEs from mice exposed to muscone compared to control mice was also somewhat surprising to us. We think that there are two potential explanations for this result: (1) Unlike subtype Olfr235, subtype Olfr1440 exhibits a significant open-side bias in newborn OSN quantities in UNO-treated adolescent females even in the absence of exposure to muscone. We speculate that this subtype (as well as subtype Olfr1431) is stimulated by odors that are emitted by female mice at the adolescent stage, and/or by another environmental source. This may limit the influence of muscone exposure on the UNO effect size. (2) There is compelling evidence that odors within the environment can enter the closed side of the OE transnasally [via the nasopharyngeal canal (Kelemen, 1947)] and/or retronasally (via the nasopharynx) in UNO-treated mice [reviewed in (Coppola, 2012)]. Thus, it is conceivable that chronic exposure of UNO-treated mice to muscone results in the eventual entry on the closed side of the OE of muscone at concentrations sufficient to promote neurogenesis. If Olfr1440 is more sensitive to muscone than Olfr235 [e.g., (Sato-Akuhara et al., 2016; Shirasu et al., 2014)], OSNs of this subtype may be especially sensitive to small amounts of odors that enter the closed side of the OE transnasally and/or retronasally. These explanations are supported by the following results:

- UNO-treated females exposed to 0.1% muscone show higher quantities of newborn Olfr1440 OSNs on both the open and closed sides of the OE in muscone exposed females compared to their unexposed counterparts (Figure 4–figure supplement 1A-middle). Similar results were also observed for newborn Olfr235 OSNs (Figure 4C-middle), albeit to a lesser extent, perhaps due to the lower sensitivity of this subtype to muscone.

- In non-occluded female mice, exposure to 0.1% muscone was found to significantly increase quantities of newborn Olfr1440 OSNs, as well as newborn Olfr235 and Olfr1431 OSNs (Figure 4D-F in revised manuscript; Figure 6 in original version). Similar results were also observed upon exposure to ambretone, another musk odor (Figure 4D-F). These experiments strongly support the hypothesis that musk odors selectively increase birthrates of OSN subtypes that they stimulate.

We have addressed these points within the results section of the revised manuscript.

(3) The authors introduce their choice to study this particular family of OSN subtypes with first, the previous finding that transcripts for one of these musk-responsive subtypes (olfr235) are downregulated in mice that are deprived of male odors. Second, musk-related odors are found in the urine of different species. This gives the misleading impression that it is known that musk-related odors are indeed excreted into male mouse urine at certain concentrations. This should be stated more clearly in the introduction (or cited, if indeed data exist that show musk-related odors in male mouse urine) because this would be a very important point from an ethological and mechanistic point of view.In addition, this would also be important information to assess if the chosen muscone concentrations fall at all into the natural range.

These are important points, which have addressed within the revised manuscript:

(1) Within the introduction, we have now stated that the emission of musk odors by mice has not been documented. We have also added extensive discussions of what is known about the emission of musk odors by mice in a new subsection within Results, as well as within the Discussion section. Most prominently, we have cited one study (Sato-Akuhara et al., 2016) that noted unpublished evidence for the emission of Olfr1440-activating compounds from male preputial glands: “Indeed, our preliminary experiments suggest that there are unidentified compounds that activate MOR215-1 in mouse preputial gland extracts.” Another study, which used histomorphology, metabolomic and transcriptomic analyses to compare the mouse preputial glands to muskrat scent glands, found that the two glands are similar in many ways, including molecular composition (Han et al., 2022). However, the study did not identify known musk compounds within mouse preputial glands.

(2) Based on the reviewer’s feedback and our own curiosity, we used GC-MS to analyze both mouse urine and preputial gland extracts for the presence of known musk odorants, particularly those known to activate Olfr235 and Olfr1440 (Sato-Akuhara et al., 2016). Although we were unable to find evidence for known musk odorants in mouse urine extracts (possibly due to insufficient sensitivity of the assay employed), we found that preputial gland extracts contain GC-MS signals that are structurally consistent with known musk odorants. A limitation of this approach, however, is that the conclusive identification of specific musk odorants in extracts derived from mouse urine and tissues requires comparisons to pure standards, many of which we could not readily obtain. For example, we were unable to obtain a pure sample of cycloheptadecanol, a musk molecule with a predicted potential match to a signal identified within preputial gland extracts. Another limitation is that although several known musk odorants have been found to activate Olfr235 and Olfr1440 OSNs, it is conceivable that structurally distinct odorants that have not yet been identified might also activate them. The findings from these experiments have been included in a new figure within the revised manuscript (Appendix 2–figure 1).

Related: If these are male-specific cues, it is interesting that changes in OR transcripts (Figure 1) can already be seen at the age of P28 where other male-specific cues are just starting to get expressed. This should be discussed.

We agree that the observed changes in quantities of newborn OSNs of musk-responsive subtypes in mice exposed to juvenile male odors deserves additional discussion. We have included a more extensive discussion of this observation in both the Results and Discussion sections of the revised manuscript.

(4) Figure 5: Under muscone exposure the number of newborn neurons on the closed sides fluctuates considerably. This doesn't seem to be the case in other experiments and raises some concerns about how reliable the naris occlusion works for strong exposure to monomolecular odors or what other potential mechanisms are at play.

We agree that the variability in quantities of newborn OSNs of musk-responsive subtypes on the closed side of the OE of UNO-treated mice deserves further discussion. As noted above, we suspect that these fluctuations are due, at least in part, to transnasal and/or retronasal odor transfer via the nasopharyngeal canal (Kelemen, 1947) and nasopharynx, respectively [reviewed in (Coppola, 2012)], which would be expected to result in exposure of the closed OE to odor concentrations that rise with increasing environmental concentrations. In support of this, quantities of newborn Olfr235 and Olfr1440 OSNs increase on both the open and closed sides with increasing muscone concentration (except at the highest concentration, 10%, in the case of Olfr1440) (Figure 4C-middle, Figure 4–figure supplement 1A-middle). It is conceivable that reductions in newborn Olfr1440 OSN quantities observed in the presence of 10% muscone reflect overstimulation-dependent reductions in survival. Our findings from UNO-based experiments are consistent with expectations that naris occlusion does not completely block exposure to odorants on the closed side, particularly at high concentrations. However, they also appear consistent with the hypothesis that exposure to musk odors promotes the neurogenesis of musk-responsive OSN subtypes.

Considering the limitations of the UNO procedure, it is important to note that the present study also includes experimental exposure of non-occluded animals to both male odors (Figure 3G, H) and exogenous musk odorants (Figures 4D-F). Findings from the latter experiments provide strong evidence that exposure to multiple musk odorants (muscone, ambretone) causes selective increases in the birthrates of multiple musk-responsive OSN subtypes (Olfr235, Olfr1440, Olfr1431).

We have included within the Results section of the revised manuscript a discussion of how observed effects of muscone exposure of UNO-treated mice may be influenced by transnasal/ retronasal odor transfer to the closed side of the OE.

(5) In contrast to all other musk-responsive OSN types, the number of newborn OSNs expressing olfr1437 increases on the closed side of the OE relative to the open in UNO-treated male mice (Figure 1). This seems to contradict the presented theory and also does not align with the bulk RNAseq data (Figure S1).

Subtype Olfr1437 is indeed an outlier among musk-responsive subtypes that were previously found to be more highly represented in the OSN population in 6-month-old sex-separated males compared to females (Appendix 1–figure 1)(C. van der Linden et al., 2018; Vihani et al., 2020). Somewhat unexpectedly, our findings from scRNA-seq experiments show slightly greater quantities of immature Olfr1437 OSNs on the closed side of the OE in juvenile males (Figure 1D, E of the revised manuscript, which now includes data from a second OE). Perhaps more informatively considering the small number of iOSNs of specific subtypes in the scRNA-seq datasets, EdU birthdating experiments show no difference in newborn Orlfr1437 OSN quantities on the 2 sides of the OE from UNO-treated juvenile males (Figure 2G). It is unclear to us why subtype Olfr1437 does not show open-side biases in newborn OSN quantities in juvenile male mice, but potential explanations include:

- Age: Findings based on bulk RNA-seq that musk responsive OSN subtypes are more highly represented in mice exposed to male odors analyzed mice that were 6 months old (C. van der Linden et al., 2018) or > 9 months old (Vihani et al., 2020) at the time of analysis. By contrast, the present study primarily analyzed mice that were juveniles (PD 28) at the time of scRNA-seq analysis (Figure 1) or EdU labeling (Figure 2G). It is conceivable that different musk-responsive subtypes are selectively responsive to distinct odors that are emitted at different ages. In this scenario, odors that increase the birthrates of Olfr235, Olfr1440, and Olfr1431 OSNs may be emitted starting at the juvenile stage, while those that increase the birthrate of Olfr1437 OSNs may be emitted in adulthood. In potential support of this, juvenile males exposed to their adult parents at the time of EdU labeling showed a slightly greater (although not statistically significantly different) UNO effect size in quantities of newborn Olfr1437 OSNs compared to controls (Figure 3–figure supplement 3).

- Capacity for stimulation-dependent neurogenesis: It is also conceivable that, unlike other musk-responsive OSN subtypes, Olfr1437 OSNs lack the capacity for stimulation-dependent neurogenesis (like the SBT-responsive subtype Olfr912, for example). If so, this would imply that increased representations of Olfr1437 OSNs observed in mice exposed to male odors for long periods (C. van der Linden et al., 2018; Vihani et al., 2020) may be due to male odor-dependent increases in the lifespans of Olfr1437 OSNs.

Within the Discussion section of the revised manuscript, we have discussed the findings concerning Olfr1437.

(6) The authors hypothesize in relation to the accelerated birthrate of musk-responsive OSN subtypes that "the acceleration of the birthrates of specific OSN subtypes could selectively enhance sensitivity to odors detected by those subtypes by increasing their representation within the OE". However, for two other OSN subtypes that detect male-specific odors, they hypothesize the opposite "By contrast, Olfr912 (Or8b48) and Olfr1295 (Or4k45), which detect the male-specific non-musk odors 2-sec-butyl-4,5-dihydrothiazole (SBT) and (methylthio)methanethiol (MTMT), respectively, exhibited lower representation and/or transcript levels in mice exposed to male odors, possibly reflecting reduced survival due to overstimulation."Without any further explanation, it is hard to comprehend why exposure to male-derived odors should, on one hand, accelerate birthrates in some OSN subtypes to potentially increase sensitivity to male odors, but on the other hand, lower transcript levels and does not accelerate birth rates of other OSN subtypes due to overstimulation.

We agree that this point deserves further explanation. Within the revised manuscript, we have expanded the Introduction and Results to describe evidence from previous studies that exposure to stimulating odors causes two categories of changes to specific OSN subtypes: elevated representations or reduced representations within the OSN population. In one study (C. J. van der Linden et al., 2020), UNO treatment was found to cause a fraction of OSN subtypes to exhibit lower birthrates and representations on the closed side of the OE relative to the open. By contrast, another fraction of OSN subtypes exhibited higher representations on the closed side of OEs from UNO-treated mice, but no difference in birthrates between the two sides. The latter subtypes were found to be distinguished by their receipt of extremely high levels of odor stimulation, suggesting that reduced odor stimulation via naris occlusion may lengthen their lifespans. In support of the possibility that Olfr912 (and Olfr1295), which detect SBT and MTMT, respectively (Vihani et al., 2020), which are emitted specifically by male mice (Lin et al., 2005; Schwende et al., 1986), UNO treatment was previously found to increase total Olfr912 OSN quantities on the closed side compared to the open side in sex-separated males (C. van der Linden et al., 2018), a finding confirmed in the present study (Figure 3–figure supplement 1H).

Taken together, findings from previous studies as well as the current one indicate that olfactory stimulation can accelerate the birthrates and/or reduced the lifespans of OSNs, depending on the specific subtypes and odors within the environment. As we have now indicated in the Discussion, we do not yet know what distinguishes subtypes that undergo stimulation-dependent neurogenesis, but it is conceivable that they detect odors with a particular salience to mice. Thus, observations that some odorants (e.g., musks) cause stimulation-dependent neurogenesis while others do not (e.g., SBT) might reflect an animal’s specific need to adapt its sensitivity to the former. Alternatively, it is conceivable that stimulation-dependent reductions in representations of subtypes such as Olfr912 and Olfr1295 reflect a fundamentally different mode of plasticity that is also adaptive, as has been hypothesized (C. van der Linden et al., 2018; Vihani et al., 2020).

**Reviewer #1 (Recommendations For The Authors):**
To support the main claim, several controls are necessary as mentioned under point 1 of the public review.As outlined in our responses to the public review, new experiments within the revised manuscript indicate the following:(1) Accelerated birthrates of 3 different musk responsive OSN subtypes (Olfr235, Olfr1440, Olfr1431) are observed in non-occluded mice following exposure to multiple exogenous musk odorants (muscone, ambretone) (Figure 4D-F).(2) Exposure of non-occluded mice to non-musk odors (SBT, IAA) does not accelerate the birthrates of musk responsive OSN subtypes (Olfr235, Olfr1440, Olfr1431) (Figure 4D-F).(3) Exposure of mice to exogenous musk odors (muscone, ambretone) does not accelerate the birthrates of non-musk responsive OSN subtypes (e.g., Olfr912), including those previously found to undergo stimulation-dependent neurogenesis (Olfr827, Olfr1325) (Figure 4–figure supplement 4C).(4) Only a fraction of OSN subtypes have a capacity to undergo accelerated neurogenesis in the presence of odors that activate them (e.g., Olfr912 birthrates are not accelerated by SBT exposure) (Figure 4–figure supplement 4C-left).In addition, this study could be considerably improved by showing that the proposed mechanism applies beyond a single OSN subtype (olfr235), especially since the most sensitive OR subtype (expressing olfr1440) does not align with the main claim. The introduction states that this is difficult because the ligands for many ORs are unknown including all subtypes previously found to undergo stimulation-dependent neurogenesis referring to your 2020 study. While this reviewer agrees that the lack of deorphanization is a significant hurdle in the field, the 2020 study states that about 4% of all ORs (which should equal >40 ORs) show a stimulus-dependent down-regulation on the closed side, not only the 7 ORs which are closer examined (Figure 1). It would tremendously improve the impact of the current study to show that the proposed effect applies also to one of these other >40 ORs.

We appreciate this question, as it alerted us to some shortcomings in how our findings were presented within the original manuscript. We respectfully disagree that only findings regarding subtype Olfr235 align with the main hypothesis of this study, which is that discrete odors can selectively promote the neurogenesis of sensory neuron subtypes that they stimulate. Specifically, we would like to draw attention to experiments on non-occluded female mice exposed to exogenous musk odorants (muscone, ambretone; revised Figures 4D-F; previously, Figure 6). Findings from these experiments provide compelling evidence that exposure to musk odorants causes selective increases in the birthrates of three different musk-responsive OSN subtypes: Olfr235, Olfr1440, and Olfr1431. Thus, we would suggest that results from the present study already show that the proposed mechanism applies to more than the just Olfr235 subtype. However, we agree with what we think is the essence of the reviewer’s point: that it is important to determine the extent to which this mechanism applies to OSN subtypes that are responsive to other (i.e., non-musk) odorants. While, as noted by the reviewer, our previous study identified several OSN subtypes that undergo stimulation-dependent neurogenesis (as well as many others that predicted to do so)(C. J. van der Linden et al., 2020), we are not aware of ligands that have been identified with high confidence for those subtypes. Although we are in the process of conducting experiments to identify additional odor/subtype pairs to which the mechanism described in this study applies, the early-stage nature of these experiments precludes their inclusion in the present manuscript.

The ethological and mechanistic relevance of the current study could be significantly improved by showing that musk-related odors that activate olfr235 are actually found in male mouse urine (and additionally are not found in female mouse urine). Otherwise, the implicated link between the acceleration of OSN birthrates by exposure to male odors and acceleration by specific monomolecular odors does not hold, raising the question of any natural relevance (e.g. the proposed adaptive function to increase sensitivity to certain odors).

As noted in our responses to the public review, we have addressed this important point within the revised manuscript as follows:

(1) We have included an extensive discussion of what is known about the emission of musk-like odors by mice.

(2) We have used GC-MS to analyze both mouse urine and preputial gland extracts for the presence of known musk compounds. Although inconclusive, we report that preputial glands contain signals that are structurally consistent with known musk compounds. The findings of these experiments have been included in the revised manuscript (new Appendix 2–figure 1), along with a discussion of their limitations.

**Reviewer #2 (Public Review):**
In their paper entitled "In mice, discrete odors can selectively promote the neurogenesis of sensory neuron subtypes that they stimulate" Hossain et al. address lifelong neurogenesis in the mouse main olfactory epithelium. The authors hypothesize that specific odorants act as neurogenic stimuli that selectively promote biased OR gene choice (and thus olfactory sensory neuron (OSN) identity). Hossain et al. employ RNA-seq and scRNA-seq analyses for subtype-specific OSN birthdating. The authors find that exposure to male and musk odors accelerates the birthrates of the respective responsive OSNs. Therefore, Hossain et al. suggest that odor experience promotes selective neurogenesis and, accordingly, OSN neurogenesis may act as a mechanism for long-term olfactory adaptation.

We appreciate this summary but would like to underscore that a mechanism involving biased OR gene choice is just one of two possibilities proposed in the Discussion section to explain how odorant stimulation of specific subtypes accelerates the birthrates of those subtypes.

The authors follow a clear experimental logic, based on sensory deprivation by unilateral naris occlusion, EdU labeling of newborn neurons, and histological analysis via OR-specific RNA-FISH. The results reveal robust effects of deprivation on newborn OSN identity. However, the major weakness of the approach is that the results could, in (possibly large) parts, depend on "downregulation" of OR subtype-specific neurogenesis, rather than (only) "upregulation" based on odor exposure. While, in Figure 6, the authors show that the observed effects are, in part, mediated by odor stimulation, it remains unclear whether deprivation plays an "active" role as well. Moreover, as shown in Figure 1C, unilateral naris occlusion has both positive and negative effects in a random subtype sample.

In our view, the present study involves two distinct and complementary experimental designs: (1) odor exposure of UNO-treated animals and (2) odor exposure of non-occluded animals. Here we address this comment with respect to each of these designs:

(1) For experiments performed on UNO-treated animals, we agree that observed differences in birthrates on the open and closed sides of the OE reflect, largely, a deceleration (i.e., downregulation) of the birthrates of these subtypes on the closed side relative to the open (as opposed to an acceleration of birthrates on the open side). Our objective in using this design was to test the extent to which specific OSN subtypes undergo stimulation-dependent neurogenesis under various odor exposure conditions. According to the main hypothesis of this study, a lower birthrate of a specific OSN subtype on the closed side of the OE compared to the open is predicted to reflect a lower level of odor stimulation on the closed side received by OSNs of that subtype. However (and as described in our responses to reviewer #1), a limitation of this design is that environmental odorants, especially at high concentrations, are likely to stimulate responsive OSNs on the closed side of the OE in addition to the open side due to transnasal and/or retronasal air flow.

(2) Experiments performed on non-occluded animals were designed to provide critical complementary evidence that specific OSN subtypes undergo accelerated neurogenesis in the presence of specific odors. Using this design, we have found compelling evidence that:

- Exposure of non-occluded mice to male odors causes the selective acceleration of the birthrate of Olfr235 OSNs (Figure 3G, H).

- Exposure of non-occluded female mice to two different musk odorants (muscone and ambretone) selectively accelerates the birthrates three different musk responsive subtypes: Olfr235, Olfr1440, and Olf1431 (Figure 4D-F and Figure 4–figure supplement 4C).

We have reorganized the revised manuscript to more clearly present the most important experimental findings using these two experimental designs. We have also highlighted (via schematics) the experimental conditions (e.g., UNO, non-occlusion, odor exposure) used for each experiment.

Another weakness is that the authors build their model (Figure 8), specifically the concept of selectivity, on a receptor-ligand pair (Olfr912 that has been shown to respond, among other odors, to the male-specific non-musk odors 2-sec-butyl-4,5-dihydrothiazole (SBT)) that would require at least some independent experimental corroboration. At least, a control experiment that uses SBT instead of muscone exposure should be performed.

We agree that this important concern deserves additional control experiments and discussion. We have addressed this concern within the revised manuscript as follows:

- Within the Results section, we have added multiple new control experiments (detailed in response to Reviewer #1), including the one recommended above. As suggested, we quantified newborn OSNs of the SBT-responsive subtype Olfr912 in non-occluded females that were either exposed to 0.1% SBT or unexposed controls. Exposure of SBT was found to cause no significant increase in quantities of newborn Olfr912 OSNs (newly added Figure 4–figure supplement 4C-left). These findings further support the model in Figure 7 (previously Figure 8) that only a fraction of OSN subtypes have a capacity to undergo accelerated neurogenesis in the presence of odors that activate them.

- Also within the Results section, we have made efforts to better highlight relevant control experiments that were included in the original version, particularly those showing that quantities of newborn Olfr912 OSNs are not affected by UNO in mice exposed to male odors (Figure 2H and Figure 3–figure supplement 1G; previously Figure 2F and Figure 3H) or by exposure of non-occluded females to male odors (Figure 3H; previously Figure 6E). Since Olfr235 is responsive to component(s) of male odors (C. van der Linden et al., 2018; Vihani et al., 2020), these results indicate that this subtype does not have the capacity of stimulation-dependent neurogenesis, which is consistent with our previous findings that only a fraction of subtypes have this capacity (C. J. van der Linden et al., 2020).

In this context, it is somewhat concerning that some results, which appear counterintuitive (e.g., lower representation and/or transcript levels of Olfr912 and Olfr1295 in mice exposed to male odors) are brushed off as "reflecting reduced survival due to overstimulation." The notion of "reduced survival" could be tested by, for example, a caspase3 assay.

This is a point that we agree deserves further discussion. Please see the explanation that we have outlined above in response to Reviewer #1.

Within the revised manuscript, we have expanded the Introduction to describe evidence from previous studies that exposure to stimulating odors causes two categories of changes to specific OSN subtypes: elevated representations or reduced representations within the OSN population. We outline evidence from previous studies that Olfr912 and Olfr1295 belong to the latter category, and that the representations of these subtypes are likely reduced by male odor overstimulation-dependent shortening of OSN lifespan.

Important analyses that need to be done to better be able to interpret the findings are to present (i) the OR+/EdU+ population of olfactory sensory neurons not just as a count per hemisection, but rather as the ratio of OR+/EdU+ cells among all EdU+ cells; and (ii) to the ratio of EdU+ cells among all nuclei (UNO versus open naris). This way, data would be normalized to (i) the overall rate of neurogenesis and (ii) any broad deprivation-dependent epithelial degeneration.

We have addressed this concern in two ways within the revised manuscript:

(1) We have noted within the Methods section that the approach of using half-sections for normalization has been used in multiple previous studies for quantifying newborn (OR+/EdU+) and total (OR+) OSN abundances (Hossain et al., 2023; Ibarra-Soria et al., 2017; C. van der Linden et al., 2018; C. J. van der Linden et al., 2020). Additionally, within the figure legends and Methods, we have more thoroughly described the approach used, including that it relies on averaging the quantifications from at least 5 high-quality coronal OE tissue sections that are evenly distributed throughout the anterior-posterior length of each OE and thereby mitigates the effects of section size and cell number variation among sections. In the case of UNO treated mice, the open and closed sides within the same section are paired, which further reduces the effects of section-to section variation. We have found that this approach yields reproducible quantities of newborn and total OSNs among biological replicate mice and enables accurate assessment of how quantities of OSNs of specific subtypes change as a result of altered olfactory experience, a key objective of this study.

(2) To assess whether the use of alternative approaches for normalizing newborn OSN quantities suggested by the reviewers would affect the present study’s findings, we compared three methods for normalizing the effects of exposure to male odors or muscone on quantities of newborn Olfr235 OSNs in the OEs of both UNO-treated and non-occluded mice: (1) OR+/EdU+ OSNs per half-section (used in this study), (2) OR+/EdU+ OSNs per total number of EdU+ cells (reviewer suggestion (i)), and (3) OR+/EdU+ OSNs per unit of DAPI+ area (an approximate measure of nuclei number; reviewer suggestion (ii)). The three normalization methods yielded statistically indistinguishable differences in assessing the effects of exposure of either UNO-treated or non-occluded mice to male odors (newly added Figure 2–figure supplement 2 and Figure 3–figure supplement 2), or of exposure of non-occluded mice to muscone (newly added Figure 4–figure supplement 3). Based on these findings, and the considerable time that would be required to renormalize all data in the manuscript, we have chosen to maintain the use of normalization per half-section.

Finally, the paper will benefit from improved data presentation and adequate statistical testing. Images in Figures 2 - 7, showing both EdU labeling of newborn neurons and OR-specific RNA-FISH, are hard to interpret. Moreover, t-tests should not be employed when data is not normally distributed (as is the case for most of their samples).

We have made extensive changes within the revised manuscript to increase the clarity and interpretability of the figures, including:

(1) Addition of a split-channel, high-magnification view of a representative image that shows the overlap of FISH and EdU signals (Figure 2D).

(2) Addition of experimental schematics and timelines corresponding to each set of experiments.

In the revised manuscript, several changes to the statistical tests have been made, as follows:

(1) To assess deviation from normality of the histological quantifications of newborn and total OSNs of specific subtypes in this study, all datasets were tested using the Shapiro-Wilk test for non-normality and the P values obtained are included in Supplementary file 1 (figure source data). Of the 274 datasets tested, 253 were found to have Shapiro-Wilk P values > 0.05, indicating that the vast majority (92%) do not show evidence of significant deviation from a normal distribution.

(2) A general lack of deviation of the datasets in this study from a normal distribution is further supported by quantile-quantile (QQ) plots, which compare actual data to a theoretically normal distribution (Appendix 4–figure 1). The datasets analyzed were separated into the following categories:

a. Quantities of newborn OSNs in UNO treated mice (Appendix 4-figure 1A)

b. Quantities of total OSNs in UNO treated mice (Appendix 4-figure 1B)

c. Quantities of newborn OSNs in non-occluded mice (Appendix 4-figure 1C)

d. UNO effect sizes for newborn or total OSNs (Appendix 4-figure 1D)

(3) Results of both parametric and non-parametric statistical tests of comparisons in this study have been included in Supplementary file 2 (statistical analyses). In general, the results from parametric and non-parametric tests are in good agreement.

(4) Statistical analyses of differences in OSN quantities in the OEs of non-occluded mice or UNO effect sizes in UNO-treated mice subjected more than two different experimental conditions have now been performed using one-way ANOVA tests, FDR-adjusted using the 2-stage linear step-up procedure of Benjamini, Krieger and Yekutieli.

**Reviewer #2 (Recommendations for the Authors):**
The manuscript by Hossain et al. would benefit from a thorough revision. Here, we outline several points that should be addressed:Figure 3E - I & Figure 4E&F: Red lines that connect mean values are misleading.

Within the revised manuscript, the UNO effect size graphs have been modified for clarity, including removal of the lines between mean values except for those comparing changes over time post EdU injection (Figure 6 and Figure 6-figure supplement 1). For these latter graphs, we think that lines help to illustrate changes in effect sizes over time.

Figure 3E - I: UNO effect sizes (right) should be tested via ANOVA.

In the revised manuscript, statistical analyses of UNO effect sizes in UNO-treated mice subjected more than two different experimental conditions were done using one-way ANOVA tests, FDR-adjusted using the 2-stage linear step-up procedure of Benjamini, Krieger and Yekutieli (Figure 2-figure supplement 2; Figure 3; Figure 3-figure supplement 1; Figure 4; Figure 4-figure supplements 1, 2). The same tests were used for analysis of differences in OSN quantities in the OEs of non-occluded mice subjected more than two different experimental conditions (Figure 3; Figure 3-figure supplement 2; Figure 4; Figure 4-figure supplements 3, 4). For comparisons of differences in quantities of newborn OSNs of musk-responsive subtypes at 4 and 7 days post-EdU between non-occluded mice exposed and unexposed to muscone, a two sample ANOVA - fixed-test, using F distribution (right-tailed) was used (Figure 6; Figure 6-figure supplement 1).

Images in Figures 2 - 7, showing both EdU labeling of newborn neurons and OR-specific RNA-FISH: Colabeling is hard / often impossible to discern. Show zoom-ins and better explain the criteria for "colabeling" in the methods.

In the revised manuscript an enlarged and split-channel view of an image showing multiple newborn Olfr235 OSNs (OR+/EdU+) has been added (Figure 2D). A detailed description of the criteria for OR+/EdU+ OSNs is provided in Methods under the section “Histological quantification of newborn and total OSNs of specific subtypes.”

Figure 1C: add Olfr912.

As a control group for iOSN quantities of musk-responsive subtypes in Figure 1, we selected random subtypes that are expressed in the same zones: 2 and 3. Olfr912 OSNs were not included because this subtype was not randomly chosen, nor is it expressed the same zones (Olfr912 is expressed in zone 4). We also note that the scRNA-seq analysis was done to allow an initial exploration of the hypothesis that some OSN subtypes with that are more highly represented in mice exposed to male odors show stimulation-dependent neurogenesis. Considering that the scRNA-seq datasets contain only small numbers of iOSNs of specific subtypes, we think they are more useful for analyzing changes in birthrates within groups of subtypes (e.g., musk responsive, random) rather than individual subtypes.

The time of OE dissection is different for data shown in Figure 1 (P28) as compared to other figures (P35). Please comment/discuss.

Within the Results section of the revised manuscript, we have now clarified that the PD 28 timepoint chosen for EdU birthdating in the histological quantification of newborn OSNs of specific subtypes is analogous to the PD 28 timepoint chosen for identification of immature (Gap43-expressing) OSNs in the scRNA-seq samples. In the case of EdU birthdating, it is necessary to provide a chase period of sufficient length to enable robust and stable expression of an OR, which defines the subtype. A chase period of 7 days was chosen based on a previous study (C. J. van der Linden et al., 2020). Hence, a dissection date of PD 35 was chosen.

Figure 3F&G: please discuss the female à female effects

In the Results and Discussion sections of the revised manuscript, we discuss our observation that the Olfr1440 and Olfr1431 subtypes show significantly higher quantities of newborn OSNs on the open side compared to closed sides in UNO-treated females. We speculate that these subtypes may receive some odor stimulation in juvenile females, perhaps via musk or related odors emitted by females themselves or from elsewhere within the environment.

Figure 4E (and other examples): male à male displays two populations (no effect versus effect); please explain/speculate.

For some UNO effect sizes, there appears to be high degree of variation among mice, and, in some cases, this diversity appears to cause the data to separate into groups. We assessed whether this diversity might reflect mice that came from different litters, but this is not the case. Rather, we speculate that the observed diversity most likely reflects low representations of newborn OSNs of some subtypes and/or under specific conditions. The data referred to by the reviewer (now Figure 3–figure supplement 3D), for example, shows UNO effect sizes for quantities of newborn Olfr1431 OSNs, which has the lowest representation among the musk-responsive subtypes analyzed in this study.

Figure 5C-E: It is unclear why strong muscone concentrations (10%) have no effect, whereas no muscone sometimes (D&E) has an effect.

As discussed in response to comments from Reviewer #1, we speculate that fluctuations in UNO effect sizes in muscone-exposed mice, particularly at high muscone concentrations, may be due, at least in part, to transnasal and/or retronasal air flow [reviewed in (Coppola, 2012)], which would be expected to result in exposure of the closed side of the OE to muscone concentrations that increase with increasing environmental concentrations. In support of this, quantities of newborn Olfr235 (Figure 4C-middle) and Olfr1440 (Figure 4–figure supplement 1A-middle) OSNs increase on both the open and closed sides with increasing muscone concentration (except at the highest concentration, 10%, in the case of Olfr1440). We speculate that reductions in newborn Olfr1440 OSN quantities observed in the presence of 10% muscone may reflect overstimulation-dependent reductions in survival.

As emphasized above, our study also includes experiments on non-occluded animals (Figures 3, 4, 5). Findings from these experiments provide additional evidence that exposure to multiple musk odorants (muscone, ambretone) causes selective increases in the birthrates of multiple musk-responsive OSN subtypes (Olfr235, Olfr1440, Olfr1431).

We have included an extensive interpretation of UNO-based experiments, including their limitations, within the Results section of the revised manuscript.

Figure S1: please explain the large error bars regarding "Transcript level".

We have clarified that the error bars in this figure, which is now Appendix 1–figure 1, correspond to 95% confidence intervals.

The figure captions could be improved for ease of reading.

Figure captions have been revised for increased clarity.

Figure 4: Include Olfr235 data for consistency.

All OSN subtypes analyzed for the effects of exposure to adult mice on UNO-induced open-side biases in quantities of newborn OSNs have been included in a single figure, which is now Figure 3–figure supplement 3.

Figure S6F&G: Do not run statistics on n = 2 (G) or 3 (F) samples.

We have removed statistical test results from comparisons involving fewer than 4 observations.

**Reviewer #3 (Public Review):**
Summary:Neurogenesis in the mammalian olfactory epithelium persists throughout the life of the animal. The process replaces damaged or dying olfactory sensory neurons. It has been tacitly that replacement of the OR subtypes is stochastic, although anecdotal evidence has suggested that this may not be the case. In this study, Santoro and colleagues systematically test this hypothesis by answering three questions: is there enrichment of specific OR subtypes associated with neurogenesis? Is the enrichment dependent on sensory stimulus? Is the enrichment the result of differential generation of the OR type or from differential cell death regulated by neural activity? The authors provide some solid evidence indicating that musk odor stimulus selectively promotes the OR types expressing the musk receptors. The evidence argues against a random selection of ORs in the regenerating neurons.Strengths:The strength of the study is a thorough and systematic investigation of the expression of multiple musk receptors with unilateral naris occlusion or under different stimulus conditions. The controls are properly performed. This study is the first to formulate the selective promotion hypothesis and the first systematic investigation to test it. The bulk of the study uses in situ hybridization and immunofluorescent staining to estimate the number of OR types. These results convincingly demonstrate the increased expression of musk receptors in response to male odor or muscone stimulation.Weaknesses:A major weakness of the current study is the single-cell RNASeq result. The authors use this piece of data as a broad survey of receptor expression in response to unilateral nasal occlusion. However, several issues with this data raise serious concerns about the quality of the experiment and the conclusions. First, the proportion of OSNs, including both the immature and mature types, constitutes only a small fraction of the total cells. In previous studies of the OSNs using the scRNASeq approach, OSNs constitute the largest cell population. It is curious why this is the case. Second, the authors did not annotate the cell types, making it difficult to assess the potential cause of this discrepancy. Third, given the small number of OSNs, it is surprising to have multiple musk receptors detected in the open side of the olfactory epithelium whereas almost none in the closed side. Since each OR type only constitutes ~0.1% of OSNs on average, the number of detected musk receptors is too high to be consistent with our current understanding and the rest of the data in the manuscript. Finally, unlike the other experiments, the authors did not describe any method details, nor was there any description of quality controls associated with the experiment. The concerns over the scRNASeq data do not diminish the value of the data presented in the bulk of the study but could be used for further analysis.

We are grateful to the reviewer for raising these important questions.

In the revised manuscript, we have clarified that the scRNA-seq dataset presented in the original version of the manuscript (now called dataset OE 1) was published and described in detail in a previous study (C. J. van der Linden et al., 2020). The reviewer is correct that the proportion of OSNs within that dataset was lower in that dataset than in other datasets that have been published more recently (using updated methods). We think this is likely because of the way that the cells were processed (e.g., from cryopreserved single cells followed by live/dead selection). However, because the open and closed sides were processed identically, we do not expect the ratios of OSNs of specific subtypes to be greatly affected. Hence, the differences observed for specific OSN subtypes on the open versus closed sides are expected to be valid.

As the reviewer notes, there is a surprisingly large difference between the number of OSNs of musk-responsive subtypes on the open and closed sides within the OE 1 dataset. This difference is a key piece of information that led us to formulate the hypothesis in the study: that musk responsive subtypes are born at a higher rate in the presence of male/musk odor stimulation. And while it is true that, on average, each subtype represents ~0.1% of the population, it is known that there is wide variance in representations among different subtypes [e.g., (Ibarra-Soria et al., 2017)]. The frequencies of the musk responsive subtypes among all OSNs on the open side of OE 1 (0.3% for Olfr235, 0.4% for olfr1440, 0.06% for Olfr1434, 0% for olfr1431, and 1% for Olfr1437) are in line with previous findings.

To confirm that the scRNA-seq findings from dataset OE 1 are not an artifact of the cell preparation methods used, we generated a second scRNA-seq dataset, OE 2, which has been added to the revised manuscript (Figure 1). The OE 2 dataset was prepared according to the same experimental timeline as OE 1, but the cells were captured immediately after dissociation and live/dead sorting via FACS. As expected, most cells within OE 2 dataset are OSNs (77% on the open side, 66% on the closed). Importantly, like the OE 1 dataset, the OE 2 dataset shows higher quantities of iOSNs of musk responsive subtypes on the open side of the OE compared to the closed (normalized for either total cells or total OSNs) (Figure 1–figure supplement 1D, E).

A weakness of the experiment assessing musk receptor expression is that the authors do not distinguish immature from mature OSNs. Immature OSNs express multiple receptor types before they commit to the expression of a single type. The experiments do not reveal whether mature OSNs maintain an elevated expression level of musk receptors.

While it is established that multiple ORs are coexpressed at a low level during OSN differentiation (Bashkirova et al., 2023; Fletcher et al., 2017; Hanchate et al., 2015; Pourmorady et al., 2024; Saraiva et al., 2015; Scholz et al., 2016; Tan et al., 2015), this has been found to occur primarily at the immediate neuronal precursor 3 (INP3) stage (Bashkirova et al., 2023; Fletcher et al., 2017), which is characterized by expression of *Tex15* (Fletcher et al., 2017; Pourmorady et al., 2024) and precedes the immature OSN (iOSN) stage, which is characterized by expression of *Gap43* (Fletcher et al., 2017; McIntyre et al., 2010; Verhaagen et al., 1989). Within the scRNA-seq datasets in the present study, iOSNs of specific subtypes are identified based on robust expression of *Gap43* (Log^2^ UMI > 1) and a specific OR gene (Log^2^ UMI > 2), as described in the figures and methods. Thus, the cells defined as iOSNs are expected to express a single OR gene and this expression should be maintained as iOSNs transition to mOSNs. To confirm these predictions, we carried out a detailed analysis of OR expression at three different stages of OSN differentiation: INP3, iOSN, and mOSN (Figure 1–figure supplement 2). The cells chosen for analysis express the musk-responsive ORs Olfr235 or Olfr1440 or a randomly chosen OR Olfr701, in addition to markers that define INP3, iOSN, or mOSN cells. As expected, individual iOSNs and mOSNs of musk-responsive subtypes were found to exhibit robust and singular OR expression on the open and closed sides of OEs from UNO-treated mice. Moreover, and as observed previously, INP3 cells coexpress multiple OR transcripts at low levels. A detailed description of how the analysis was performed is included in the Methods section under Quantification and statistical analysis.

Within the histology-based quantifications, newborn OSNs are identified based on their robust RNA-FISH signals corresponding to a specific OR transcript and an EdU label. Considering the EdU chase time of 7 days, most EdU-positive cells are expected to have passed the INP3 stage and be iOSNs or mOSNs. Moreover, considering the low level of OR expression within INP3 cells, it is unlikely OR transcripts are expressed at a high enough level to be detectable and/or counted at this stage and thereby affect newborn OSN quantifications.

There are also two conceptual issues that are of concern. The first is the concept of selective neurogenesis. The data show an increased expression of musk receptors in response to male odor stimulation. The authors argue that this indicates selective neurogenesis of the musk receptor types. However, it is not clear what the distinction is between elevated receptor expression and a commitment to a specific fate at an early stage of development. As immature OSNs express multiple receptors, a likely scenario is that some newly differentiated immature OSNs have elevated expression of not only the musk receptors but also other receptors. The current experiments do not distinguish the two alternatives. Moreover, as pointed out above, it is not clear whether mature OSNs maintain the increased expression. Although a scRNASeq experiment can clarify it, the authors, unfortunately, did not perform an in-depth analysis to determine at which point of neurogenesis the cells commit to a specific musk receptor type. The quality of the scRNASeq data unfortunately also does not lend confidence for this type of analysis.

The addition of a second scRNA-seq dataset within the revised manuscript (Figure 1), combined with the new scRNA-seq-based analyses of OR expression in INP3, iOSN, and mOSN cells (Figure 1-figure supplement 2), provide strong evidence that iOSNs and mOSNs robustly express a single OR gene and that cellular expression is stable from the iOSN to the mOSN stage. These analyses do not support a scenario in which odor stimulation causes upregulated expression of multiple ORs and thereby causes apparent increases in quantities of newly generated OSNs that express musk-responsive ORs. Rather, the data firmly support a mechanism in which odor stimulation increases quantities of newly generated OSNs that have stably committed to the robust expression of a single musk-responsive OR.

A second conceptual issue, the idea of homeostasis in regeneration, which the authors presented in the Introduction, needs clarification. In its current form, it is confusing. It could mean that a maintenance of the distribution of receptor types, or it could mean the proper replacement of a specific OR type upon the loss of this type. The authors seem to refer to the latter and should define it properly.

We have revised the Introduction section to clarify our use of the term homeostatic in one instance (paragraph 4) and replace it with more specific language in a second instance (paragraph 5).

**Reviewer #3 (Recommendations For The Authors):**
Concerns over scRNASeq data. It appears that the samples may have included non-OE tissues, which reduced the representation of the OSNs. This experiment may need to be repeated to increase the number of OSNs.

As outlined in the response to the public comments, we think that the low proportion of OSNs in the OE 1 data set reflects how the cells were prepared and processed. We have now included a second scRNA-seq dataset to address this concern.

Cell types should be identified in the scRNASeq analysis, and the number of cells documented for each cell type, at least for the OSNs. The data should be made available for general access.

We have now clarified that the OE 1 dataset was published as part of a previous study (C. J. van der Linden et al., 2020) and was made publicly available as part of that study (https://www.ncbi.nlm.nih.gov/geo/query/acc.cgi?acc=GSE157119). All cell types in the newly generated OE 2 dataset have been annotated (Figure 1) and this dataset has also been made publicly available (https://www.ncbi.nlm.nih.gov/geo/query/acc.cgi?acc=GSE278693). The numbers and percentages of OSNs within OE 1 and OE 2 datasets have been added to the legend of Figure 1-figure supplement 1.

The specific OR types should be segregated for mature and immature OSNs. The percentage of a specific OR type should be normalized to the total number of OSNs, rather than the total cells. The current quantification is misleading because it gives the false sense that the muscone receptors represent ~0.1% of cells when the proportion is much higher if only OSNs are considered.

In the revised manuscript, quantities of iOSNs (*Gap43*+ cells) of specific subtypes within the OE 1 and OE 2 scRNA-seq datasets are graphed as percentages of both all OSNs (Figure 1E, Figure 1–figure supplement 1D) and all cells (Figure 1–figure supplement 1E). As a percentage of all OSNs, average quantities of iOSNs of musk responsive subtypes on the open side of the OE range from 0.005% (for Olfr1431) to 0.14% (for Olfr1440) (Figure 1E).

Within the feature plots for the two datasets, the differentiation stages of indicated OSNs have been clearly defined within the figures and figure legends. For the OE 1 dataset, iOSNs are differentiated from mOSNs by arrows (Figure 1–figure supplement 1C). For the OE 2 dataset (Figure 1D), only immature OSNs are shown for simplicity.

Technical details of the scRNASeq should be documented. In the feature plot of musk-response receptors (Figure. 1D), it is better to use the actual quantity of expression rather than binarized representation (with or without an OR). If one needs to use on/off to determine the number of cells for a given OR type, then the criteria of selection should be given.

Technical details of generation of the scRNA-seq datasets have been documented in the “Method details” section (for the OE 2 dataset) and in the method section of our previous publication of the OE 1 dataset (C. J. van der Linden et al., 2020). Details of the scRNA-seq analyses, including the criteria used to define immature OSNs of specific subtypes, are documented within the “Quantification and statistical analysis” section.

Within the feature plots, we have decided to show OSNs of a given subtype in a binary fashion using specific colors for the sake of simplicity (Figure 1D, Figure 1-figure supplement 1C). To address the reviewer’s cooncern, we have added a new figure that provides detailed information about OR transcript expression (levels and genes) within iOSNs and mOSNs of two different musk responsive subtypes and a randomly chosen subtype (Figure 1-figure supplement 2).

An in-depth analysis of the onset of OR expression in the GBC, INP, immature, and mature OSNs should be performed. It is also important to determine how many other receptors are detected in the cells that express the musk receptors. The current scRNASeq data may not be of sufficiently high quality and the experiment needs to be repeated. It is also important for the authors to take measures to eliminate ambient RNA contamination.

The revised manuscript includes a second scRNA-seq dataset (OE 2; Figure 1). Details of how both the original (OE 1) and new datasets were generated have been documented within the Methods sections of the corresponding publications [(C. J. van der Linden et al., 2020); present study]. For both datasets, live/dead selection of cells was performed, which was expected to reduce ambient RNA.

The revised manuscript also includes a new figure that provides detailed information about OR transcript expression within INP3, iOSN and mOSN cells that express one of two different musk responsive ORs or a randomly chosen OR (Figure 1-figure supplement 2). These data reveal, as reported previously (Bashkirova et al., 2023; Fletcher et al., 2017; Pourmorady et al., 2024), that low levels of multiple OR transcripts are detected in INP3 (*Tex15*+) cells. By contrast, iOSN (Gap43+) and mOSN (Omp+) cells robustly express a single OR, with little or no expression of other ORs.

Quantification of cells for Figure 2-7 should be changed. Instead of using cell number per 1/2 section, the data should be calculated using density using the area of the epithelium or normalized to the total number of cells (based on DAPI staining). This is because multiple sections are taken from the same mouse along the A-P axis. These sections have different sizes and numbers of cells.

As noted in response to a similar concern of Reviewer #2, this has been addressed in two ways within the revised manuscript:

(1) We have noted within the Methods section that the approach of using half-sections for normalization has been used in multiple previous studies for quantifying newborn (OR+/EdU+) and total (OR+) OSN abundances (Hossain et al., 2023; Ibarra-Soria et al., 2017; C. van der Linden et al., 2018; C. J. van der Linden et al., 2020). Additionally, within the figure legends and Methods, we have more thoroughly described the approach used, including that it relies on averaging the quantifications from at least 5 high-quality coronal OE tissue sections that are evenly distributed throughout the anterior-posterior length of each OE and thereby mitigates the effects of section size and cell number variation among sections. In the case of UNO treated mice, the open and closed sides within the same section are paired, which further reduces the effects of section-to section variation. We have found that this approach yields reproducible quantities of newborn and total OSNs among biological replicate mice and enables accurate assessment of how quantities of OSNs of specific subtypes change as a result of altered olfactory experience, a key objective of this study.

(2) To assess whether the use of alternative approaches for normalizing newborn OSN quantities suggested by the reviewers would affect the present study’s findings, we compared three methods for normalizing the effects of exposure to male odors or muscone on quantities of newborn Olfr235 OSNs in the OEs of both UNO-treated and non-occluded mice: (1) OR+/EdU+ OSNs per half-section (used in this study), (2) OR+/EdU+ OSNs per total number of EdU+ cells (reviewer suggestion (i)), and (3) OR+/EdU+ OSNs per unit of DAPI+ area (an approximate measure of nuclei number; reviewer suggestion (ii)). The three normalization methods yielded statistically indistinguishable differences in assessing the effects of exposure of either UNO-treated or non-occluded mice to male odors (newly added Figure 2–figure supplement 2 and Figure 3–figure supplement 2), or of exposure of non-occluded mice to muscone (newly added Figure 4–figure supplement 3). Based on these findings, and the considerable time that would be required to renormalize all data in the manuscript, we have chosen to maintain the use of normalization per half-section.

References

Bashkirova, E. V., Klimpert, N., Monahan, K., Campbell, C. E., Osinski, J., Tan, L., Schieren, I., Pourmorady, A., Stecky, B., Barnea, G., Xie, X. S., Abdus-Saboor, I., Shykind, B. M., Marlin, B. J., Gronostajski, R. M., Fleischmann, A., & Lomvardas, S. (2023). Opposing, spatially-determined epigenetic forces impose restrictions on stochastic olfactory receptor choice. eLife, 12, RP87445. https://doi.org/10.7554/eLife.87445

Coppola, D. M. (2012). Studies of olfactory system neural plasticity: The contribution of the unilateral naris occlusion technique. Neural Plasticity, 2012, 351752. https://doi.org/10.1155/2012/351752

Fletcher, R. B., Das, D., Gadye, L., Street, K. N., Baudhuin, A., Wagner, A., Cole, M. B., Flores, Q., Choi, Y. G., Yosef, N., Purdom, E., Dudoit, S., Risso, D., & Ngai, J. (2017). Deconstructing Olfactory Stem Cell Trajectories at Single-Cell Resolution. Cell Stem Cell, 20(6), 817-830.e8. https://doi.org/10.1016/j.stem.2017.04.003

Han, X., Jiang, Y., Feng, N., Yang, P., Zhang, M., Jin, W., Zhang, T., Huang, Z., Zhao, H., Zhang, K., Liu, S., & Hu, D. (2022). Comparison of the Homology Between Muskrat Scented Gland and Mouse Preputial Gland. Journal of Mammalian Evolution, 29(2), 435–446. https://doi.org/10.1007/s10914-022-09604-w

Hanchate, N. K., Kondoh, K., Lu, Z., Kuang, D., Ye, X., Qiu, X., Pachter, L., Trapnell, C., & Buck, L. B. (2015). Single-cell transcriptomics reveals receptor transformations during olfactory neurogenesis. Science (New York, N.Y.), 350(6265), 1251–1255. https://doi.org/10.1126/science.aad2456

Hossain, K., Smith, M., & Santoro, S. W. (2023). A histological protocol for quantifying the birthrates of specific subtypes of olfactory sensory neurons in mice. STAR Protocols, 4(3), 102432. https://doi.org/10.1016/j.xpro.2023.102432

Ibarra-Soria, X., Nakahara, T. S., Lilue, J., Jiang, Y., Trimmer, C., Souza, M. A., Netto, P. H., Ikegami, K., Murphy, N. R., Kusma, M., Kirton, A., Saraiva, L. R., Keane, T. M., Matsunami, H., Mainland, J., Papes, F., & Logan, D. W. (2017). Variation in olfactory neuron repertoires is genetically controlled and environmentally modulated. eLife, 6. https://doi.org/10.7554/eLife.21476

Kelemen, G. (1947). The junction of the nasal cavity and the pharyngeal tube in the rat. Archives of Otolaryngology, 45(2), 159–168. https://doi.org/10.1001/archotol.1947.00690010168002

Lin, D. Y., Zhang, S.-Z., Block, E., & Katz, L. C. (2005). Encoding social signals in the mouse main olfactory bulb. Nature, 434(7032), 470–477. https://doi.org/10.1038/nature03414

McIntyre, J. C., Titlow, W. B., & McClintock, T. S. (2010). Axon growth and guidance genes identify nascent, immature, and mature olfactory sensory neurons. Journal of Neuroscience Research, 88(15), 3243–3256. https://doi.org/10.1002/jnr.22497

Pourmorady, A. D., Bashkirova, E. V., Chiariello, A. M., Belagzhal, H., Kodra, A., Duffié, R., Kahiapo, J., Monahan, K., Pulupa, J., Schieren, I., Osterhoudt, A., Dekker, J., Nicodemi, M., & Lomvardas, S. (2024). RNA-mediated symmetry breaking enables singular olfactory receptor choice. Nature, 625(7993), 181–188. https://doi.org/10.1038/s41586-023-06845-4

Saraiva, L. R., Ibarra-Soria, X., Khan, M., Omura, M., Scialdone, A., Mombaerts, P., Marioni, J. C., & Logan, D. W. (2015). Hierarchical deconstruction of mouse olfactory sensory neurons: From whole mucosa to single-cell RNA-seq. Scientific Reports, 5, 18178. https://doi.org/10.1038/srep18178

Sato-Akuhara, N., Horio, N., Kato-Namba, A., Yoshikawa, K., Niimura, Y., Ihara, S., Shirasu, M., & Touhara, K. (2016). Ligand Specificity and Evolution of Mammalian Musk Odor Receptors: Effect of Single Receptor Deletion on Odor Detection. The Journal of Neuroscience: The Official Journal of the Society for Neuroscience, 36(16), 4482–4491. https://doi.org/10.1523/JNEUROSCI.3259-15.2016

Scholz, P., Kalbe, B., Jansen, F., Altmueller, J., Becker, C., Mohrhardt, J., Schreiner, B., Gisselmann, G., Hatt, H., & Osterloh, S. (2016). Transcriptome Analysis of Murine Olfactory Sensory Neurons during Development Using Single Cell RNA-Seq. Chemical Senses, 41(4), 313–323. https://doi.org/10.1093/chemse/bjw003

Schwende, F. J., Wiesler, D., Jorgenson, J. W., Carmack, M., & Novotny, M. (1986). Urinary volatile constituents of the house mouse,*Mus musculus*, and their endocrine dependency. Journal of Chemical Ecology, 12(1), 277–296. https://doi.org/10.1007/BF01045611

Shirasu, M., Yoshikawa, K., Takai, Y., Nakashima, A., Takeuchi, H., Sakano, H., & Touhara, K. (2014). Olfactory receptor and neural pathway responsible for highly selective sensing of musk odors. Neuron, 81(1), 165–178. https://doi.org/10.1016/j.neuron.2013.10.021

Tan, L., Li, Q., & Xie, X. S. (2015). Olfactory sensory neurons transiently express multiple olfactory receptors during development. Molecular Systems Biology, 11(12), 844. https://doi.org/10.15252/msb.20156639

van der Linden, C. J., Gupta, P., Bhuiya, A. I., Riddick, K. R., Hossain, K., & Santoro, S. W. (2020). Olfactory Stimulation Regulates the Birth of Neurons That Express Specific Odorant Receptors. Cell Reports, 33(1), 108210. https://doi.org/10.1016/j.celrep.2020.108210

van der Linden, C., Jakob, S., Gupta, P., Dulac, C., & Santoro, S. W. (2018). Sex separation induces differences in the olfactory sensory receptor repertoires of male and female mice. Nature Communications, 9(1), 5081. https://doi.org/10.1038/s41467-018-07120-1

Verhaagen, J., Oestreicher, A. B., Gispen, W. H., & Margolis, F. L. (1989). The expression of the growth associated protein B50/GAP43 in the olfactory system of neonatal and adult rats. The Journal of Neuroscience: The Official Journal of the Society for Neuroscience, 9(2), 683–691.

Vihani, A., Hu, X. S., Gundala, S., Koyama, S., Block, E., & Matsunami, H. (2020). Semiochemical responsive olfactory sensory neurons are sexually dimorphic and plastic. eLife, 9, e54501. https://doi.org/10.7554/eLife.54501